# AMD: Angular Margin based Knowledge Distillation

## Abstract

Knowledge distillation as a broad class of methods has led to the development of lightweight and memory efficient models, using a pre-trained model with a large capacity (teacher network) to train a smaller model (student network). Recently, additional variations for knowledge distillation, utilizing activation maps of intermediate layers as the source of knowledge, have been studied. Generally, in computer vision applications it is seen that the feature activation learned by a higher-capacity model contains richer knowledge, highlighting complete objects while focusing less on the background. Based on this observation, we leverage the teacher's dual ability to accurately distinguish between positive (relevant to the target object) and negative (irrelevant) areas. We propose a new type of distillation, called angular margin-based distillation (AMD). AMD uses the angular distance between positive and negative features by projecting them onto a hypersphere, motivated by the near angular distributions seen in many feature extractors. Then, we create a more attentive feature from encoded knowledge by the angular distance by introducing an angular margin to the positive feature. Transferring such knowledge from the teacher network enables the student model to harness the teacher's better discrimination of positive and negative features, thus distilling superior student models. The proposed method is evaluated for various student-teacher network pairs on three public datasets. Furthermore, we show that the proposed method has advantages in compatibility with other learning techniques, such as using fine-grained features, augmentation, and other distillation methods.

## 1 Introduction

In the past decade, convolutional neural networks (CNN) have been widely deployed into many commercial applications. Various architectures that go beyond convolutional methods have also been developed. However, a core challenge in all of them is that they are accompanied by high computational complexity, and large storage requirements (Gou et al., 2021; Cho & Hariharan, 2019). For this reason, application of deep networks is still limited to environments that have massive computational support. In emerging applications, there is growing demand for applying deep nets on edge, mobile, and Iot devices (Li et al., 2018; Plastiras et al., 2018; Jang et al., 2020; Wu et al., 2016). To move beyond these limitations, many studies have developed a lightweight form of neural models which assure performance while 'lightening' the network scale (Li et al., 2018; Plastiras et al., 2018; Cho & Hariharan, 2019; Jang et al., 2020; Han et al., 2016; Hinton et al., 2015; Wu et al., 2016).

Knowledge distillation (KD) is one of the promising solutions that can reduce the network size and develop an efficient network model (Cho & Hariharan, 2019; Yim et al., 2017; Gou et al., 2021). The concept of knowledge distillation is that the network consists of two networks, a larger one called teacher and a smaller one called student (Hinton et al., 2015). During training the student, the teacher transfers its knowledge to the student, using the logits from the final layer. So, the student can retain the teacher model's classification performance.

Recently, feature-based distillation methods for KD have been studied to learn richer information from the teacher for better-mimicking and performance improvement (Gou et al., 2021; Wang & Yoon, 2021). Romero et al. (2015) firstly introduced the use of intermediate representations in FitNets using feature-based distillation. This method enables the student to mimic the teacher's feature maps in intermediate layers. In attention transfer (Zagoruyko & Komodakis, 2017; Wang et al., 2020b; Ji et al., 2021; Gou et al., 2021) which is one of the popular methods for feature-based distillation, spatial attention maps are used as the source of knowledge for distillation with intermediate layers, where the maps are computed by summation of the squared activations and represent where it concentrates. The method encourages the student to generate similar normalized maps as the teacher. However, these studies have

only focused on mimicking the teacher's activation from a layer (Wang & Yoon, 2021), not considering the teacher's dual ability to accurately distinguish between positive (relevant to the target object) and negative (irrelevant). The emphasized positive feature regions that encapsulate regions of the target object are crucial to predicting the correct class. In general, a higher-capacity model shows better performance, producing those regions with more attention and precision compared to the smaller network. This gives an insight that the transfer of distinct regions of the positive and negative pairs from teacher to student could significantly improve performance.

In this paper, based on this motivation, we propose an angular margin based distillation (AMD), that is motivated by training discriminative angular distance on a hypersphere manifold. Recent insights have shown that features learnt in deep-networks often exhibit an angular distribution, usually leveraged via a hyperspherical embedding (Choi et al., 2020; Liu et al., 2016; 2017). Such embeddings lead to improved discriminative power, and feature separability. In terms of loss-functions these can be implemented by using angular features that correspond to the geodesic distance on the hypersphere and incorporating a preset constant margin. In this work, we show that leveraging such spherical embeddings also improves knowledge distillation. Firstly, to get more activated features, spatial attention maps are computed and decoupled into two parts: positive and negative maps. Secondly, we construct a new form of knowledge by projecting the features onto the hypersphere to reflect the angular distance between them. Then, we introduce an angular margin to the positive feature to get a more attentive representation of the feature. Finally, during the distillation, the student tries to mimic the more separated decision regions of the teacher to improve the classification performance. Therefore, the proposed method trains the student model effectively.

The contributions of this paper are:

- We propose a new knowledge distillation, called angular margin distillation (AMD), using the angular distance of attentive features on the hypersphere.
- We experimentally show that the proposed method results in significant improvements with different combinations of networks and also outperforms other attention based methods across three datasets having different complexities.
- We corroborate results from previous studies which suggest that the performance of a higher capacity teacher model is not necessarily better.
- We rigorously validate the advantages of the proposed distillation method with various aspects using visualization of activation maps, classification accuracy, and reliability diagrams.

The rest of the paper is organized as follows. In section 2, we describe related work. In section 4, we provide an overview of the proposed method. In section 5, we describe our experimental results and analysis. In section 6, we discuss our findings and conclusions.

## 2 Related Work

**Knowledge Distillation.** Knowledge distillation, a transfer learning method, trains a smaller model by shifting knowledge from a larger model. KD is firstly introduced by Buciluǎ et al. (2006) and is further explored by Hinton et al. (2015). The main concept of KD is using soft labels by a trained teacher network. That is, mimicking soft probabilities helps students get knowledge of teachers, which improves beyond using hard labels (training labels) alone. Cho & Hariharan (2019) explore which combination of student-teacher is good to obtain the better performance. They show that using a teacher trained by early stopping the training improves the efficacy of KD. KD can be categorized into two approaches that use the outputs of the teacher (Gou et al., 2021). One is response-based KD, which uses the posterior probabilities with softmax loss. The other is feature-based KD using the intermediate features with normalization. Feature-based methods can be performed with the response-based method to complement traditional KD (Gou et al., 2021).

**Attention Transfer.** To capture the better knowledge of a teacher network, Zagoruyko & Komodakis (2017) suggest activation-based attention transfer (AT), which uses a sum of squared attention mapping function computing statistics across the channel dimension. Although the depth of teacher and student is different, knowledge can be transferred by the attention mapping function, which matches the depth size as one. The activation-based spatial attention maps are created as: $f_{sum}^d(A) = \sum_{j=1}^{c} |A_j|^d$, where $f$ is a computed attention map, $A$ is an output of a layer, $c$ is the number of channels for the output, $j$ is the number for the channel, and $d > 1$. A higher value of $d$ corresponds to a heavier

weight on the most discriminative parts defined by activation level. AT (feature-based distillation method) shows better effectiveness when used with traditional KD (response-based KD) (Zagoruyko & Komodakis, 2017).

**Spherical Feature Embeddings.** The majority of existing methods (Sun et al., 2014; Wen et al., 2016) rely on Euclidean distance for feature distinction. These approaches could not solve the problem that classification under open-set protocol shows a meaningful result only when successfully narrowing maximal intra-class distance. To solve this problem, an angular-softmax (A-softmax) function is proposed to distinguish the features by increasing the angular margins between features (Liu et al., 2017). According to its geometric interpretation, using A-softmax function equivalents to the projection of features onto the hypersphere manifold, which intrinsically matches the preliminary condition that features also lie on a manifold. Applying the angular margin penalty corresponds to the geodesic distance margin penalty in the hypersphere (Liu et al., 2017). A-softmax function encourages learned features to be discriminative on hypersphere manifold. For this reason, the A-softmax function shows superior performance to the original softmax function when tested on several classification problems (Liu et al., 2017). On the other hand, Choi et al. (2020) introduced angular margin based contrastive loss (AMC-loss) as an auxiliary loss, employing the discriminative angular distance metric that corresponds to geodesic distance on a hypersphere manifold. AMC-loss increases inter-class separability and intra-class compactness, improving performance in classification. The method can be combined with other deep techniques, because it easily encodes the angular distributions obtained from many types of deep feature learners (Choi et al., 2020).

## 3 Background

### 3.1 Traditional Knowledge Distillation

In standard knowledge distillation (Hinton et al., 2015), the loss for training a student is:

$$\mathcal{L} = (1 - \lambda)\mathcal{L}_\mathcal{C} + \lambda\mathcal{L}_\mathcal{K}, \tag{1}$$

where, $\mathcal{L}_\mathcal{C}$ denotes the standard cross entropy loss, $\mathcal{L}_\mathcal{K}$ is KD loss, and $\lambda$ is a hyperparameter; $0 < \lambda < 1$. The error between the output of the softmax layer of a student network and the ground-truth label is penalized by the cross-entropy loss:

$$\mathcal{L}_\mathcal{C} = \mathcal{H}(softmax(a_S), y), \tag{2}$$

where $\mathcal{H}(\cdot)$ is a cross entropy loss function, $a_S$ is the logits of a student (inputs to the final softmax), and $y$ is a ground truth label. The outputs of student and teacher are matched by KL-divergence loss:

$$\mathcal{L}_\mathcal{K} = \tau^2 KL(z_T, z_S), \tag{3}$$

where, $z_T = softmax(a_T/\tau)$ is a softened output of a teacher network, $z_S = softmax(a_S/\tau)$ is a softened output of a student, and $\tau$ is a hyperparameter; $\tau > 1$. Feature distillation methods using intermediate layers can be used with the standard knowledge distillation that uses output logits. When they are used together, in general, it is beneficial to guide the student network towards inducing more similar patterns of teachers and getting a better classification performance. Thus, we also utilize the standard knowledge distillation with our proposed method.

### 3.2 Attention Map

Denote an output map as $A \in \mathbb{R}^{c \times h \times w}$, where $c$ is the number of output channels, $h$ is the height for the size of output, and $w$ is width for the size of the output. The attention map for the teacher is given as follows:

$$f_T^l = \sum_{j=1}^{c} |A_{T,j}^l|^2 \tag{4}$$

Here, $A_T$ is an output of a layer from a teacher, $l$ is a specific layer, $c$ is the number of channels, $j$ is the number for the output channel, and $T$ denotes a teacher network. The attention map for the student is $f_S^{l'} = \sum_{j'=1}^{c'} |A_{S,j'}^{l'}|^2$, where $A_S$ is an output of a layer from a student, $l'$ is the corresponding layer of $l$, $c'$ is the number of channels for the output, $j'$ is the number for the output channel, and $S$ denotes a student network. If the student and teacher use the same depth for transfer, $l'$ can be the layer at the same depth as $l$; if not, $l'$ can be the end of the same block for the teacher. From the attention map, we obtain positive and negative maps and we project features onto hypersphere to calculate angular distance for distillation. The details are explained in section 4.

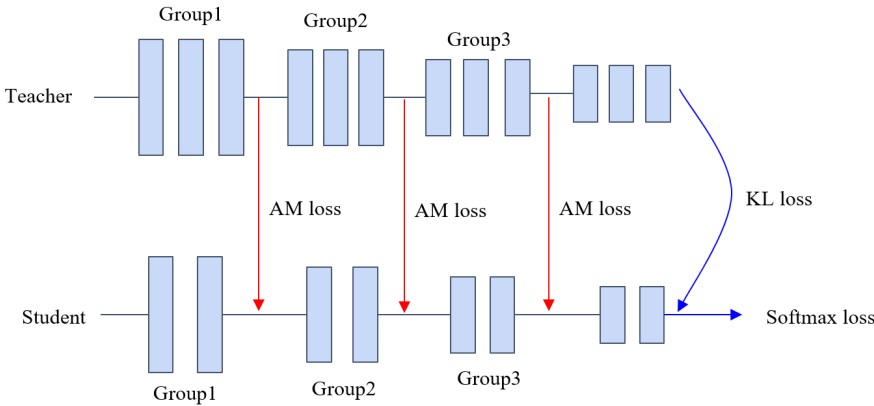

Figure 1: Schematics of teacher-student knowledge transfer with the proposed method

### 3.3 Spherical Feature with Angular Margin

In order to promote the learned features to have an angular distribution, (Liu et al., 2017; Wang et al., 2018a) proposed to introduce the angular distance between features $W$ and weights $x$. For example, $W^T x = \|W\|\|x\|cos(\theta)$, where bias is set as 0 for simplicity, and $\theta$ is the angle between $W$ and $x$. Then, the normalization of feature and weight makes the outputs only depend on the angle between weights and features and further, $\|x\|$ is replaced to a constant $s$ such that the features are distributed on a hypersphere with a radius of $s$. To enhance the discrimination power, angular margin $m$ is applied to the angle of the target. Finally, ouput logits are used to formulate probability with angular margin $m$ as below (Liu et al., 2017; Wang et al., 2018a):

$$G^i = log\left(\frac{e^{s\cdot(cos(m\cdot\theta_{y_i}))}}{e^{s\cdot(cos(m\cdot\theta_{y_i}))} + \sum_{j=1,j\neq y_i}^{J} e^{s\cdot(cos(\theta_j))}}\right), \tag{5}$$

where $y_i$ is a label and $\theta_{y_i}$ is a target angle for class $i$, $\theta_j$ is an angle obtained from $j$-th element of output logits, and $J$ is the class number. Liu et al. (2017) and Wang et al. (2018a) utilized output logits to obtain more discriminative features for classification on a hypersphere manifold, which performs better than using original softmax function. We adopt Equation 5 to create the new type of feature-knowledge in the intermediate layers instead of output logits in the final classifier, thereby more attentive feature maps are transferred to the student model.

## 4 Proposed Method

The overall approach for the proposed method is illustrated in Figure 1. The proposed method utilizes features from intermediate layers. The approach for extracting AM knowledge is illustrated in Figure 2. To obtain the angular distance between positive and negative features, first, we generate attention maps from outputs of intermediate layers. And then, we obtain positive and negative features and calculate probabilities for distillation. The details for obtaining the positive and negative maps and angular margin based knowledge are explained follows.

### 4.1 Generating Attention Maps

To transfer activated features from teacher to student, the output of intermediate layers are used. To match the dimension size between teacher and student models, we create the normalized attention maps (Zagoruyko & Komodakis, 2017), which has benefits in generating maps discriminatively between positive and negative features. This reduces the need for any additional training procedure for matching the channel dimension sizes between teacher and student. We use the power value $d = 2$ for generating the attention maps, which shows the best results as reported in previous methods (Zagoruyko & Komodakis, 2017).

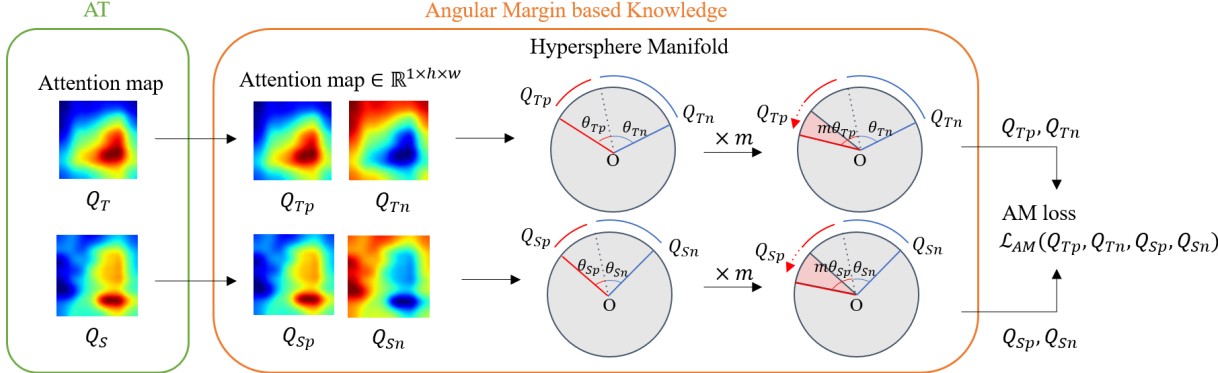

Figure 2: Schematics of teacher-student AM knowledge transfer. Based on the attention maps, positive and negative maps are obtained. Based on their angle features, angular probability is calculated. To enhance the discrimination power, angular margin $m$ is multiplied to the positive angle. After then, angular margin based knowledge is obtained and go through to AM loss function to transfer the knowledge. The details are explained in section 4.2.

## 4.2 Angular Margin Computation

Although the activation map-based distillation provides additional context information for student model learning, there is still room to craft an attentive activation map that can distill a superior student model in KD. To further refine the original attention map, we propose an angular margin-based distillation (AMD) that encodes new knowledge using the angular distance between positive (relevant to the target object) and negative features (irrelevant) on the hypersphere.

In this section, the details for extracting the proposed knowledge using positive and negative features are explained. For a clear notation, we write the normalized positive map as $Q_p = f/\|f\|$ where $f$ is the output map extracted from the intermediate layer in networks. Further, we can obtain the normalized negative map by $Q_n = 1 - Q_p$.

Then, to make the positive map more attentive, we insert an angular margin $m$ into the positive features. In this way, a new feature-knowledge encoding attentive feature can be defined as follows:

$$G^l(Q_p, Q_n) = log\left(\frac{e^{s\cdot(cos(m\theta_{p_l}))}}{e^{s\cdot(cos(m\theta_{p_l}))} + e^{s\cdot(cos(\theta_{n_l}))}}\right),\qquad(6)$$

where $\theta_{p_l} = cos^{-1}(Q_p)$ and $\theta_{n_l} = cos^{-1}(Q_n)$ for $l^{th}$ layer in the networks, and $m$ is a scalar angular margin. $G^l \in \mathbb{R}^{1\times h\times w}$ reflects the angular distance between positive and negative features in $l^{th}$ layer. For transferring knowledge, we aim to make the student's $G^l(Q_{Sp}, Q_{Sn})$ approximate the teacher's $G^l(Q_{Tp}, Q_{Tn})$ by decreasing the angular distance between feature maps.

## 4.3 Angular Margin based Distillation Loss

With redesigned knowledge encoding terms as above, we finally define the angular margin distillation (AMD) loss that accounts for the knowledge gap between the teacher and student activations as:

$$\mathcal{L}_{AM}(Q_{Tp}, Q_{Tn}, Q_{Sp}, Q_{Sn}) = \frac{1}{3|L|}\sum_{(l,l')\in L}\left(\underbrace{\left\|\hat{G}^l(Q_{Tp}, Q_{Tn}) - \hat{G}^{l'}(Q_{Sp}, Q_{Sn})\right\|_F^2}_{\mathbf{A}} + \underbrace{\left\|\hat{Q}_{Tp}^l - \hat{Q}_{Sp}^{l'}\right\|_F^2}_{\mathbf{P}} + \underbrace{\left\|\hat{Q}_{Tn}^l - \hat{Q}_{Sn}^{l'}\right\|_F^2}_{\mathbf{N}}\right),\quad(7)$$

Here, $\hat{G}$ denotes a function for normalization for output of function G, $\hat{Q}$ is a normalized map. $L$ collects the layer pairs ($l$ and $l'$), and $\|\cdot\|_F$ is the Frobenius norm (Tung & Mori, 2019). This criterion encourages the student to learn more attentive features from the teacher network. We will verify the performance of each component (A, P, and N) in Section 5.2.

The final loss ($\mathcal{L}_{AMD}$) of our proposed method combines all the distillation losses, including the conventional logit distillation (Equation 3). Thus, our overall learning objective can be written as:

$$\mathcal{L}_{AMD} = \lambda_1 \mathcal{L}_{\mathcal{C}} + \lambda_2 \mathcal{L}_{\mathcal{K}} + \gamma \mathcal{L}_{\mathcal{A}}, \tag{8}$$

where $\mathcal{L}_{\mathcal{C}}$ is a cross-entropy loss, $\mathcal{L}_{\mathcal{K}}$ is a knowledge distillation loss, $\mathcal{L}_{\mathcal{A}}$ denotes the angular margin based loss from $\mathcal{L}_{AM}$, and $\lambda_1$, $\lambda_2$, and $\gamma$ are hyperparameters to control the balance between different losses.

**Global and Local Feature Distillation.** So far, we only consider the global feature (i.e., preserving its dimension and size). However, we point out that the global feature sometimes does not transfer more informative knowledge and rich spatial information across contexts of an input. Therefore, we also suggest utilizing local features during distillation. Specifically, the global feature is the original feature without a map division. Local features are determined by the division of the global feature. We split the global feature map from each layer by 2 for the width and height sizes of the maps to create four ($2 \times 2$) local feature maps. That is, one local map has $h/2 \times w/2$ size, where $h$ and $w$ are the height and width sizes of the global map. Similar to before, local features encoding the attentive angle can be extracted for both teacher and student. Then, the losses considering global and local features for our method are:

$$\mathcal{L}_{\mathcal{A}_{\text{global}}} = \mathcal{L}_{AM}(Q_T, Q_S), \ \ \mathcal{L}_{\mathcal{A}_{\text{local}}} = \frac{1}{K} \sum_{k=1}^{K} \mathcal{L}_{AM}(Q_T^k, Q_S^k), \tag{9}$$

where $Q_T$ and $Q_S$ are global features of the teacher and student for distillation, and $Q_T^k$ and $Q_S^k$ are local features of the teacher and student, respectively, for $k$-th element of $K$, where $K$ is the total number of local maps from a map; $K = 4$. When $\mathcal{L}_{\mathcal{A}_{\text{global}}}$ and $\mathcal{L}_{\mathcal{A}_{\text{local}}}$ are used together, we applied weights of 0.2 for local and 0.8 for global features to make a balance for learning.

## 5 Experiments

In this section, we present experimental validation of the proposed method. We evaluate the proposed method, AMD, with various combinations of teacher and student, which have different architectural styles. We run experiments on three public datasets that have different complexities. We examine the sensitivity with several different hyperparameters ($\gamma$ and $m$) for the proposed distillation and discuss which setting is the best. To demonstrate the detailed contribution, we report the results with various aspects, using classification accuracy as well as activation maps extracted by Grad-CAM (Selvaraju et al., 2017). Finally, we investigate performance enhancement by combining previous methods including filtered feature based distillation.

### 5.1 Dataset Description and Experimental Settings

#### 5.1.1 Dataset

**CIFAR-10.** CIFAR-10 dataset (Krizhevsky & Hinton, 2009) includes 10 classes with 5000 training images per class and 1000 testing images per class. Each image is an RGB image of size $32 \times 32$. We use the 50000 images as the training set and 10000 as the testing set. The experiments on CIFAR-10 helps validate the efficacy of our models with less time consumption.

**CINIC-10.** We extend our experiments on CINIC-10 (Darlow et al., 2018). CINIC-10 comprises of augmented extension in the style of CIFAR-10, but the dataset contains 270,000 images whose scale is closer to that of ImageNet. The images are equally split into each 'train', 'test', and 'validate' sets. The size of the images is $32 \times 32$. There are ten classes with 9000 images per class.

**Tiny-ImageNet / ImageNet.** To extend our experiments on a larger scale dataset having more complexity, we use Tiny-ImageNet (Le & Yang, 2015). The size of the images for Tiny-ImageNet is $64 \times 64$. We pad them to $68 \times 68$, then they are randomly cropped to $64 \times 64$, and horizontally flipped, for augmentation to account for the complexity of the dataset. The training and testing sets are of size 100k and 10k respectively. The dataset includes 200 classes. For ImageNet (Deng et al., 2009), The dataset has 1k categories with 1.2M training images. The images are randomly cropped and then resized to $224 \times 224$ and horizontally flipped.

### 5.1.2 Settings for experiments

Table 1: Architecture of WRN used in experiments. Down-sampling is performed in the first layers of conv3 and conv4. 16 and 28 mean depth and $k$ is width (channel multiplication) of the network.

| Group Name | Output Size | WRN16-$k$ | WRN28-$k$ |
|---|---|---|---|
| conv1 | 32×32 | 3×3, 16 | 3×3, 16 |
| conv2 | 32×32 | $\begin{bmatrix} 3\times3,\ 16k \\ 3\times3,\ 16k \end{bmatrix}\times2$ | $\begin{bmatrix} 3\times3,\ 16k \\ 3\times3,\ 16k \end{bmatrix}\times4$ |
| conv3 | 16×16 | $\begin{bmatrix} 3\times3,\ 32k \\ 3\times3,\ 32k \end{bmatrix}\times2$ | $\begin{bmatrix} 3\times3,\ 32k \\ 3\times3,\ 32k \end{bmatrix}\times4$ |
| conv4 | 8×8 | $\begin{bmatrix} 3\times3,\ 64k \\ 3\times3,\ 64k \end{bmatrix}\times2$ | $\begin{bmatrix} 3\times3,\ 64k \\ 3\times3,\ 64k \end{bmatrix}\times4$ |
|  | 1×1 | average pool, 10-d fc, softmax | |

For experiments on CIFAR-10, CINIC-10, and Tiny-ImageNet, we set the batch size as 128, the total epochs as 200 using SGD with momentum 0.9, a weight decay of $1 \times 10^{-4}$, and the initial learning rate $lr$ as 0.1 which is decayed by a factor of 0.2 at epochs 40, 80, 120, and 160. For ImageNet, we use SGD with momentum of 0.9 and the batch size is set as 256. We run a total epoch of 100. The initial learning rate $lr$ is 0.1 decayed by 0.1 in 30, 60, and 90 epochs.

In experiments, we use the proposed method with WideResNet (WRN) (Zagoruyko & Komodakis, 2016) for teacher and student models to evaluate the classification accuracy, which is popularly used for KD (Cho & Hariharan, 2019; Zagoruyko & Komodakis, 2017; Yim et al., 2017; Tung & Mori, 2019). Their network architectures are described in Table 1.

To determine optimal parameters $\lambda_1$ and $\lambda_2$ for KD, we tested with different values for $\lambda_1$ and $\lambda_2$ for training based on KD on CIFAR-10 dataset. As shown in Figure 3, when $\lambda_1$ is 0.1 and $\lambda_2$ is 0.9 ($\tau = 4$) with KD, the accuracy of a student (WRN16-1) trained with WRN16-3 as a teacher is the best. If $\lambda_1$ is small and $\lambda_2$ is large, the distillation effect of KD is increased. Since the accuracy depends on $\lambda_1$ and $\lambda_2$, we referred to previous studies (Cho & Hariharan, 2019; Tung & Mori, 2019; Ji et al., 2021) to choose the popular parameters for experiments. The parameters of ($\lambda_1 = 0.1$, $\lambda_2 = 0.9$, $\tau = 4$), ($\lambda_1 = 0.4$, $\lambda_2 = 0.6$, $\tau = 16$), ($\lambda_1 = 0.7$, $\lambda_2 = 0.3$, $\tau = 16$), and ($\lambda_1 = 1.0$, $\lambda_2 = 1.0$, $\tau = 4$) are used for KD on CIFAR-10, CINIC-10, Tiny-ImageNet, and ImageNet, respectively.

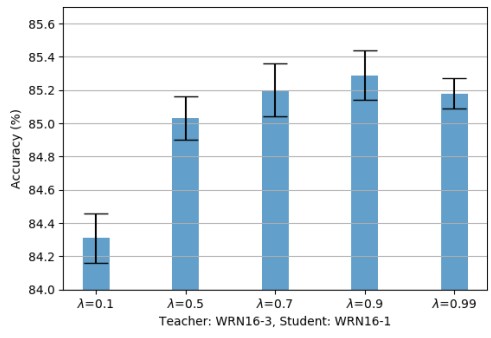

Figure 3: Accuracy (%) of students (WRN16-1) trained with a teacher (WRN16-3) on CIFAR-10 for various $\lambda_2$. $\lambda_1$ is obtained by 1 - $\lambda_2$.

We perform baseline comparisons with traditional KD (Hinton et al., 2015), attention transfer (AT) (Zagoruyko & Komodakis, 2017), relational knowledge distillation (RKD) (Park et al., 2019), variational information distillation (VID) (Ahn et al., 2019), similarity-preserving knowledge distillation (SP) (Tung & Mori, 2019), correlation congruence for knowledge distillation (CC) (Peng et al., 2019), contrastive representation distillation (CRD) (Tian et al., 2019), attentive feature distillation and selection (AFDS) (Wang et al., 2020b), and attention-based feature distillation (AFD) (Ji et al., 2021) that is a new feature linking method considering similarities between the teacher and student features, including state-of-the-art approaches. Note, the distillation methods are performed with traditional KD to see if they enhance standard KD, keeping the same setting as the proposed method. The constant parameter $s$ and margin parameter $m$ for the proposed method are 64 and 1.35, respectively. The loss weight $\gamma$ of the proposed method is 5000. We determine the hyperparameters empirically, considering the distillation effects by the capacity of models. A more detailed description of parameters appears in section 5.4. All experiments were repeated five times, and the averaged best accuracy and the standard deviation of performance are reported.

No augmentation method is applied for CIFAR-10 and CINIC-10. For the proposed method, additional techniques, such as using the other hidden layers for generating better distillation effects from teachers or reshaping the dimension size of the feature maps, are not applied. All of our experiments are run on a 3.50 GHz CPU (Intel® Xeon(R) CPU E5-1650 v3), 48 GB memory, and NVIDIA TITAN Xp (3840 NVIDIA® CUDA® cores and 12 GB memory) graphic card (NVIDIA, 2016).

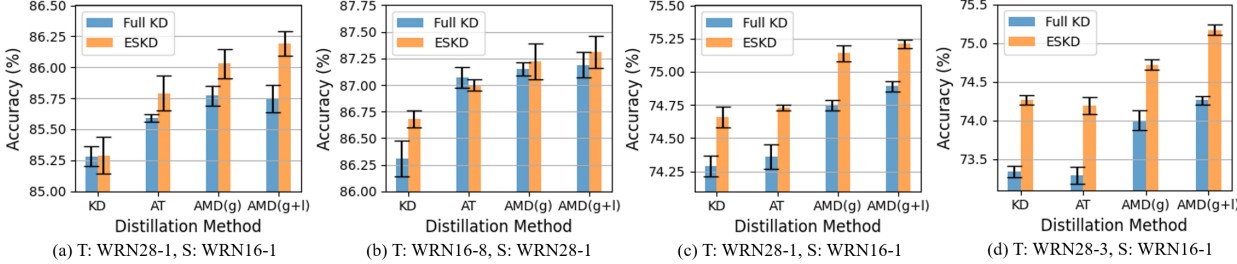

(a) T: WRN28-1, S: WRN16-1    (b) T: WRN16-8, S: WRN28-1    (c) T: WRN28-1, S: WRN16-1    (d) T: WRN28-3, S: WRN16-1

Figure 4: Accuracy (%) for Full KD and ESKD. (a) and (b) are on CIFAR-10, and (c) and (d) are on CINIC-10, respectively. T and S denotes teacher and student models, respectively.

To obtain the best performance, we adopt early-stopped KD (ESKD) (Cho & Hariharan, 2019) for training teacher and student models, leveraging its effects across the board in improving the efficacy of knowledge distillation. As shown in Figure 4, the early stopped model of a teacher tends to train student models better than Full KD that uses a fully trained teacher.

## 5.2 Attention based Distillation

Table 2: Details of teacher and student network architectures. ResNet (He et al., 2016) and WideResNet (Zagoruyko & Komodakis, 2016) are denoted by ResNet (depth) and WRN (depth)-(channel multiplication), respectively.

| Setup | Compression type | Teacher | Student | # of params (teacher) | # of params (student) | Compression ratio |
|---|---|---|---|---|---|---|
| (a) | Channel | WRN16-3 | WRN16-1 | 1.5M | 0.18M | 11.30% |
| (b) | Depth | WRN28-1 | WRN16-1 | 0.37M | 0.18M | 47.38% |
| (c) | Depth+Channel | WRN16-3 | WRN28-1 | 1.5M | 0.37M | 23.85% |
| (d) | Different architecture | ResNet44 | WRN16-1 | 0.66M | 0.18M | 26.47% |

In this section, we explore the performance of attention based distillation approaches with different types of combinations for teacher and student. We set four types of combinations for teacher and student that consist of the same or different structure of networks. The four types of combinations are described in Table 2. Since the proposed method is relevant to using attention maps, we implemented various baselines that are state-of-the-art attention based distillation methods, including AT (Zagoruyko & Komodakis, 2017), AFDS (Wang et al., 2020b), and AFD (Ji et al., 2021). As described in section 2, AT (Zagoruyko & Komodakis, 2017) uses activation-based spatial attention maps for transferring from teacher to student. AFDS (Wang et al., 2020b) includes attentive feature distillation and accelerates the transfer-learned model by feature selection. Additional layers are used to calculate a transfer importance predictor used to measure the importance of the source activation maps and enforce a different penalty for training a student. AFD (Ji et al., 2021) extracts channel and spatial attention maps and identifies similar features between teacher and

Table 3: Accuracy (%) on CIFAR-10 with various knowledge distillation methods. The methods denoted by "*" are attention based distillation. "g" and "l" denote using global and local feature distillation, respectively.

| Setup | Method | | | | | | | | | |
|---|---|---|---|---|---|---|---|---|---|---|
| | Teacher | Student | KD | AT* | SP | RKD | VID | AFDS* | AFD* | AMD (g) | (g+l) |
| (a) | 87.76 ±0.12 | 84.11 ±0.12 | 85.29 ±0.15 | 85.79 ±0.14 | 85.69 ±0.11 | 85.45 ±0.09 | 85.40 ±0.14 | _ | 86.23 ±0.13 | 86.28 ±0.06 | **86.36** ±0.10 |
| (b) | 85.59 ±0.13 | 84.11 ±0.12 | 85.48 ±0.12 | 85.79 ±0.12 | 85.77 ±0.07 | 85.47 ±0.12 | 84.92 ±0.13 | 85.53 ±0.13 | 85.84 ±0.11 | 86.04 ±0.12 | **86.10** ±0.10 |
| (c) | 87.76 ±0.12 | 85.59 ±0.12 | 86.57 ±0.16 | 86.77 ±0.11 | 86.56 ±0.09 | 86.38 ±0.22 | 86.64 ±0.24 | _ | 87.24 ±0.03 | 87.13 ±0.14 | **87.35** ±0.10 |
| (d) | 86.41 ±0.20 | 84.11 ±0.21 | 85.44 ±0.06 | 85.95 ±0.05 | 85.41 ±0.12 | 85.50 ±0.06 | 85.17 ±0.11 | 85.14 ±0.13 | 85.78 ±0.09 | 86.22 ±0.07 | **86.34** ±0.05 |

Table 4: Accuracy (%) on CINIC-10 with various knowledge distillation methods. The methods denoted by "*" are attention based distillation. AMD outperforms RKD (Park et al., 2019). "g" and "l" denote using global and local feature distillation, respectively. $c^a$ setup consists of WRN28-3 teacher and WRN16-1 student with compression ratio of 5.31%.

| Setup | Teacher | Student | Method KD | AT* | SP | VID | AFDS* | AFD* | AMD (g) | AMD (g+l) |
|---|---|---|---|---|---|---|---|---|---|---|
| (a) | 75.40 ±0.12 | | 74.31 ±0.10 | 74.63 ±0.13 | 74.43 ±0.14 | 74.35 ±0.05 | − | 74.13 ±0.12 | 75.04 ±0.11 | **75.18** ±0.09 |
| (b) | 75.59 ±0.15 | 72.05 ±0.12 | 74.66 ±0.08 | 74.73 ±0.02 | 74.94 ±0.11 | 73.85 ±0.08 | 74.54 ±0.08 | 74.36 ±0.04 | 75.14 ±0.06 | **75.21** ±0.04 |
| ($c^a$) | 76.97 ±0.05 | | 74.26 ±0.06 | 74.19 ±0.11 | 75.05 ±0.10 | 74.06 ±0.15 | − | 74.20 ±0.12 | 74.72 ±0.07 | **75.17** ±0.07 |
| (d) | 74.30 ±0.15 | | 74.47 ±0.09 | 74.67 ±0.05 | 74.46 ±0.17 | 74.43 ±0.10 | 74.64 ±0.12 | 73.31 ±0.13 | 74.93 ±0.07 | **75.10** ±0.10 |

Table 5: Accuracy (%) on Tiny-ImageNet with various knowledge distillation methods. The methods denoted by "*" are attention based distillation. AMD outperforms VID (Ahn et al., 2019) and RKD (Park et al., 2019). "g" and "l" denote using global and local feature distillation, respectively. $b^b$ setup consists of WRN40-1 teacher and WRN16-1 student with compression ratio of 32.53%. $c^b$ setup comprises of WRN40-2 teacher and WRN16-1 student with compression ratio of 8.27%.

| Setup | Teacher | Student | Method KD | AT* | SP | AFDS* | AFD* | AMD (g) | AMD (g+l) |
|---|---|---|---|---|---|---|---|---|---|
| (a) | 58.16 ±0.30 | | 49.99 ±0.15 | 49.72 ±0.15 | 49.27 ±0.19 | − | 50.00 ±0.23 | **50.32** ±0.07 | 49.92 ±0.04 |
| ($b^b$) | 54.74 ±0.24 | 49.45 ±0.20 | 49.56 ±0.17 | 49.79 ±0.22 | 49.89 ±0.20 | 49.46 ±0.28 | 50.04 ±0.27 | **50.15** ±0.10 | 49.97 ±0.18 |
| ($c^b$) | 59.92 ±0.15 | | 49.67 ±0.13 | 49.62 ±0.16 | 49.59 ±0.25 | − | 49.78 ±0.24 | 49.88 ±0.20 | **50.07** ±0.10 |
| (d) | 54.66 ±0.14 | | 49.52 ±0.16 | 49.45 ±0.28 | 49.13 ±0.20 | 49.55 ±0.13 | 49.44 ±0.27 | 49.92 ±0.09 | **50.08** ±0.16 |

Table 6: Top-1 and Top-5 accuracy (%) on ImageNet with various knowledge distillation methods. The methods denoted by "*" are attention based distillation. "g" and "l" denote using global and local feature distillation, respectively.

| | Teacher | Student | KD | AT* | RKD | SP | CC | AFD* | CRD(+KD) | AMD (g) | AMD (g+l) |
|---|---|---|---|---|---|---|---|---|---|---|---|
| Top-1 | 73.31 | 69.75 | 70.66 | 70.70 | 70.59 | 70.79 | 69.96 | 71.38 | 71.17(71.38) | **71.58** | 71.47 |
| Top-5 | 91.42 | 89.07 | 89.88 | 90.00 | 89.68 | 89.80 | 89.17 | − | 90.13(90.49) | **90.50** | 90.49 |

student, which are used to control the distillation intensities for all possible pairs and compensate for the limitation of learning to transfer (L2T) (Jang et al., 2019) using manually selected links. We implemented AFDS (Wang et al., 2020b) when the dimension size of features for intermediate layers from the student is the same as the one from the teacher to concentrate on the distillation effects. We use four datasets that have varying degrees of difficulty in a classification problem. These baselines are used in the following experiments as well.

Table 3 presents the accuracy of various knowledge distillation methods for all setups in Table 2 on CIFAR-10 dataset. The proposed method, AMD (global+local), has the best performing results in all cases. Table 4 describes the CINIC-10 results. In most cases, AMD (global+local) achieves the best results. For experiments on Tiny-ImageNet, as illustrated in Table 5, AMD outperforms previous methods, and AMD (global) shows better results in (a) and ($b^b$) setups. For ($c^b$) and (d) setups, AMD (global+local) provides better results. For experiments on ImageNet, standard KD is not applied to baselines and Full KD is utilized. Teacher and student networks are ResNet34 and ResNet18,

respectively. The results of baselines are referred from prior works (Tian et al., 2019; Ji et al., 2021). As described in Table 6, AMD (global) outperforms other distillation methods, increasing the top-1 and top-5 accuracy by 1.83% and 1.43% over the results of learning from scratch, respectively.

Compared to KD, AT obtains better performance in most cases across datasets. That is, the attention map helps the teacher to transfer its knowledge. Even though there is a case that AT shows lower performance than KD in Table 5, AMD outperforms KD in all cases. It verifies that applying the discriminative angular distance metric for knowledge distillation maximizes the attention map's efficacy of transferring the knowledge and performs to complement the traditional KD for various combinations of teacher and student. The accuracies of SP with setup (a) and (d), and AFD with setup (d), are even lower than the accuracy of learning from scratch, while AMD performs better than other methods as shown in Table 5. When the classification problem is harder, AMD (global) can perform better than AMD (global+local) in some cases. When the teacher and student have different channels or architectural styles, AMD (global+local) can generate a better student than AMD (global).

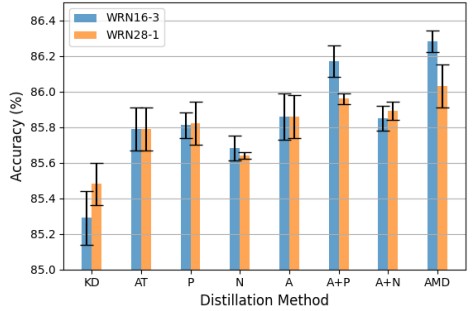

Figure 5: Accuracy (%) of students (WRN16-1) trained with teachers (WRN16-3 and WRN28-1) on CIFAR-10 for various loss functions.

**Components of AMD Loss Function.** As described in Equation 7, angular margin distillation loss function ($\mathcal{L}_{AM}(Q_{Tp}, Q_{Tn}, Q_{Sp}, Q_{Sn})$) incldues three components (A, P, N). To verify the performance of each component in AMD loss, we experiment with each component separately. As shown in Figure 5, among all components, A provides the strongest contribution. Also, the combination of all the components (AMD) shows a much higher performance. This result indicates that all components (AMD) are critical to distilling the best student model.

By using KD, SP, and AMD (global), Figure 6 plots $\mathcal{L}_A$ vs. accuracy for WRN16-1 students trained with WRN16-3, WRN28-1, and ResNet44 teachers, on CIFAR-10 testing set. As shown in Figure 6, when the loss value is smaller, the accuracy is higher. Thus, these plots verify that $\mathcal{L}_A$ and performance are correlated.

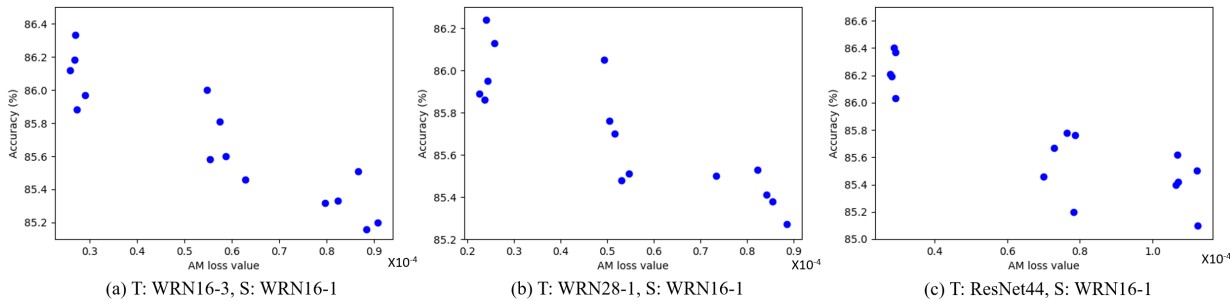

(a) T: WRN16-3, S: WRN16-1     (b) T: WRN28-1, S: WRN16-1     (c) T: ResNet44, S: WRN16-1

Figure 6: $\mathcal{L}_A$ vs. Accuracy (%) for (from left to right) WRN16-1 students (S) trained with WRN16-3, WRN28-1, and ResNet44 teachers (T), on CIFAR-10.

## 5.3 Various Capacity of Teachers

To understand the effect of the capacity of the teacher, we implemented various combinations of teacher and student, where the teacher has a different capacity/ We use well-known benchmarks for image classification which are WRN (Zagoruyko & Komodakis, 2016), ResNet (He et al., 2016), and MobileNetV2 (M.NetV2) (Sandler et al., 2018). We applied the same settings as in the experiments of the previous section.

The results in classification accuracy for the student models are described in Table 7 across three datasets, trained with attention based and non-attention based methods (Hinton et al., 2015; Zagoruyko & Komodakis, 2017; Tung & Mori, 2019). The number of trainable parameters are noted in in brackets. For all cases, the proposed method, AMD, shows the highest accuracy. When the complexity of the dataset is higher and the depth of teacher is largely different from the

Table 7: Accuracy (%) with various knowledge distillation methods for different combinations of teachers and students. "Teacher" and "Student" denote results of the model used to train the distillation methods and trained from scratch, respectively. "g" and "l" denote using global and local feature distillation, respectively.

| Method | CIFAR-10 | | | | CINIC-10 | | | | | | | | | Tiny-ImageNet | | |
|---|---|---|---|---|---|---|---|---|---|---|---|---|---|---|---|---|
| Teacher | WRN 28-1 (0.4M, 85.84) | WRN 40-1 (0.6M, 86.39) | WRN 16-3 (1.5M, 88.15) | WRN 16-8 (11.0M, 89.50) | WRN 16-3 (1.5M, 75.65) | WRN 16-8 (11.0M, 77.97) | WRN 28-1 (0.4M, 73.91) | WRN 40-1 (0.6M, 74.49) | WRN 28-3 (3.3M, 77.14) | WRN 40-2 (2.2M, 76.66) | WRN 16-3 (1.5M, 75.65) | WRN 28-3 (3.3M, 77.14) | M.Net V2 (0.6M, 80.98) | WRN 40-1 (0.6M, 55.28) | WRN 40-2 (2.3M, 60.18) | WRN 16-3 (1.6M, 58.78) |
| Student | WRN16-1 (0.2M, 84.11±0.21) | | WRN28-1 (0.4M, 85.59±0.13) | | WRN16-1 (0.2M, 72.05±0.12) | | | | | | ResNet20 (0.3M, 72.74±0.09) | | | WRN16-1 (0.2M, 49.45±0.20) | | ResNet20 (0.3M, 51.75±0.19) |
| KD | 85.48 ±0.12 | 85.42 ±0.11 | 86.57 ±0.16 | 86.68 ±0.08 | 74.31 ±0.10 | 74.17 ±0.16 | 74.66 ±0.08 | 74.45 ±0.03 | 74.26 ±0.06 | 74.29 ±0.09 | 75.12 ±0.11 | 74.97 ±0.07 | 76.69 ±0.06 | 49.56 ±0.17 | 49.67 ±0.13 | 51.72 ±0.13 |
| AT | 85.79 ±0.12 | 85.79 ±0.11 | 86.77 ±0.11 | 87.00 ±0.05 | 74.63 ±0.13 | 74.23 ±0.14 | 74.73 ±0.02 | 74.55 ±0.06 | 74.19 ±0.11 | 74.48 ±0.08 | 75.33 ±0.11 | 75.18 ±0.09 | 77.34 ±0.10 | 49.79 ±0.22 | 49.62 ±0.16 | 51.65 ±0.05 |
| SP | 85.77 ±0.07 | 85.90 ±0.11 | 86.56 ±0.09 | 86.94 ±0.08 | 74.43 ±0.11 | 74.34 ±0.13 | 74.94 ±0.11 | 74.86 ±0.07 | 75.04 ±0.10 | 74.81 ±0.09 | 75.29 ±0.10 | 75.50 ±0.09 | 73.71 ±0.10 | 49.89 ±0.20 | 49.59 ±0.25 | 51.87 ±0.09 |
| AMD (g) | 86.04 ±0.12 | 86.03 ±0.09 | 87.13 ±0.14 | 87.22 ±0.17 | 75.04 ±0.11 | 74.93 ±0.09 | 75.14 ±0.06 | **75.12** ±0.07 | 74.72 ±0.07 | 74.95 ±0.20 | 75.66 ±0.08 | 75.61 ±0.06 | 78.45 ±0.03 | **50.15** ±0.11 | 49.88 ±0.20 | 51.89 ±0.25 |
| AMD (g+l) | **86.10** ±0.10 | **86.15** ±0.06 | **87.35** ±0.10 | **87.31** ±0.15 | **75.18** ±0.09 | **75.20** ±0.05 | **75.21** ±0.04 | 75.10 ±0.04 | **75.22** ±0.07 | **75.04** ±0.06 | **75.75** ±0.08 | **75.76** ±0.11 | **78.62** ±0.04 | 49.97 ±0.18 | **50.07** ±0.10 | **52.12** ±0.15 |

one of the student, AMD (global) tends to generate a better student than AMD (global+local). When a larger capacity of students is used, the accuracy observed is higher. This is seen in the results from WRN16-1 and ResNet20 students with WRN16-3 and WRN28-3 teachers on CINIC-10 dataset. For the combinations, ResNet20 students having a larger capacity than WRN16-1 generate better results. Furthermore, on CIFAR-10, when a WRN16-3 teacher is used, a WRN28-1 student achieves 87.35% for AMD (global+local), whereas a WRN16-1 student achieves 86.36% for AMD (global+local). On Tiny-ImageNet, when AMD (global+local) is used, the accuracy of a ResNet20 student is 52.12%, which is higher than the accuracy of a WRN16-1 student, which is 49.92%.

Compared to KD, in most cases, AT achieves better performance. However, when the classification problem is difficult, such as when using Tiny-ImageNet, and when WRN40-2 teacher and WRN16-1 student are used, both AT and SP show worse performance than KD. When the WRN16-3 teacher and ResNet20 student are used, KD and AT perform worse than the model trained from scratch. The result of AT is even lower than that of KD. So, there are cases where AT and SP cannot complement the performance of the traditional KD. On the other hand, for the proposed method, the results are better than the baselines in all the cases. Interestingly, on CIFAR-10 and CINIC-10, the result of a WRN16-1 student trained by AMD with a WRN28-1 teacher is even better than the result of the teacher. Therefore, we conclude that the proposed method maximizes the attention map's efficacy of transferring the knowledge and complements traditional KD.

Also, when applying the larger teacher model and the smaller student model, the performance degradation of AMD can occur. For example, on CINIC-10, WRN16-1 student trained with WRN40-1 (0.6M) teacher outperforms the one trained with WRN40-2 (2.3M) teacher. Both AMD and other methods produce some cases with lower performance when a better (usually larger) teacher is used. These findings support previous research (Cho & Hariharan, 2019; Stanton et al., 2021; Wang & Yoon, 2021) that a better teacher does not always guarantee a better student.

## 5.4 Sensitivity Analysis

In this section, we investigate sensitivity for hyperparameters ($\gamma$ and $m$) used for the angular margin based attention distillation.

### 5.4.1 Effect of angular distillation hyperparameter $\gamma$

The results of a student model (WRN16-1) for AMD (global) trained with teachers (WRN16-3 and WRN28-1) by using various $\gamma$ on CIFAR-10 (the first row) and CINIC-10 (the second row) are depicted in Figure 7 ($m = 1.35$). When $\gamma$ is 5000, all results show the best accuracy. For CIFAR-10, when WRN16-3 is used as a teacher, the accuracy of $\gamma = 3000$ is higher than that of $\gamma = 7000$. However, for WRN28-1 as a teacher, the accuracy of $\gamma = 7000$ is higher than that of $\gamma = 3000$. When $\gamma$ is 1000, the accuracy is lower than KD, implying that it does not complement KD and adversely affects the performance. On the other hand, for CINIC-10, when the WRN16-3 teacher is used, the result

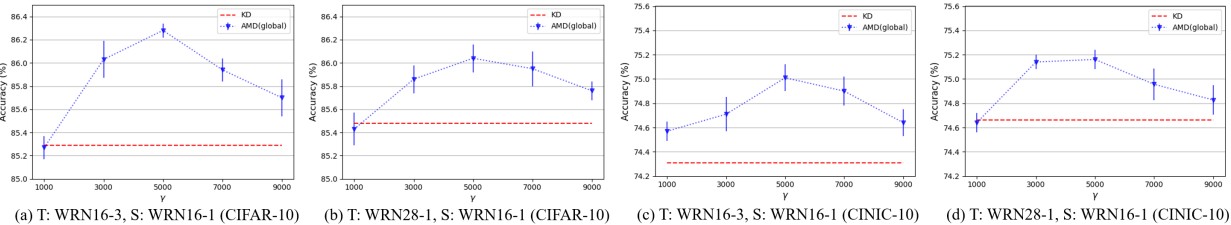

Figure 7: Accuracy (%) of students (WRN16-1) for AMD (global) with various $\gamma$, trained with teachers (WRN16-3 and WRN28-1) on CIFAR-10 and CINIC-10. "T" and "S" denote teacher and student, respectively.

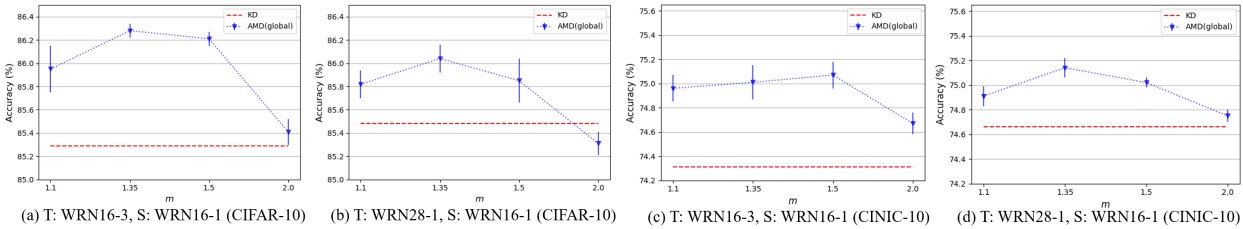

Figure 8: Accuracy (%) of students (WRN16-1) for AMD (global) with various angular margin $m$, trained with teachers (WRN16-3 and WRN28-1) on CIFAR-10 and CINIC-10. "T" and "S" denote teacher and student, respectively.

of $\gamma = 7000$ is better than that of $\gamma = 3000$. But, for the WRN28-1 teacher, $\gamma = 3000$ is higher than that of $\gamma = 7000$. Therefore, $\gamma$ values between 3000 and 7000 achieve good performance, while too small or large $\gamma$ values do not help much with improvement. Therefore, setting the proper $\gamma$ value is important for performance. We recommend using $\gamma$ as 5000, which produces the best results across datasets and combinations of teacher and student.

### 5.4.2 Analysis of angular margin $m$

The results of a student model (WRN16-1) for AMD (global) trained with teachers (WRN16-3 and WRN28-1) by various angular margin $m$ on CIFAR-10 (the first row) and CINIC-10 (the second row) are illustrated in Figure 8 ($\gamma = 5000$). As described in section 4.2, using the large value of $m$ corresponds to producing more distinct positive features in the attention map and making a large gap between positive and negative features for distillation. When $m$ is 1.35 for the WRN16-3 teacher, the WRN16-1 student shows the best performance of 86.28% on CIFAR-10. When $m = 1.5$ for CINIC-10, the student's accuracy is 75.13%, which is higher than when $m = 1.35$. When the teacher is WRN28-1, the student produces the best accuracy with $m = 1.35$ on both datasets. The student model with $m = 1.35$ performs better than the one with $m = 1.1$ and 2.0. When the complexity of the dataset is higher, using $m$ (1.5) which is larger than 1.35 can produce a good performance. When $m = 1.0$ (no additional margin applied to the positive feature) for CIFAR-10 and CINIC-10 with setup ($b$), the results are 85.81% and 74.83%, which are better than those of 85.31% and 74.75% from $m = 2.0$, respectively. This result indicates that it is important to set an appropriate $m$ value for our method. We believe that angular margin plays a key role in determining the gap between positive and negative features. As angular margin increases, the positive features are further emphasized, and in this case of over-emphasis by a much larger $m$, the performance is worse than that of the smaller $m$. We recommend using a margin $m$ of around 1.35 ($m > 1.0$), which generates the best results in most cases.

### 5.5 Analysis with Activation Maps

To analyze results with intermediate layers, we adopt Grad-CAM (Selvaraju et al., 2017) which uses class-specific gradient information to visualize the coarse localization map of the important regions in the image. In this section, we present the activation maps from intermediate layers and the high level of the layer with various methods. The red region is more crucial for the model prediction than the blue one.

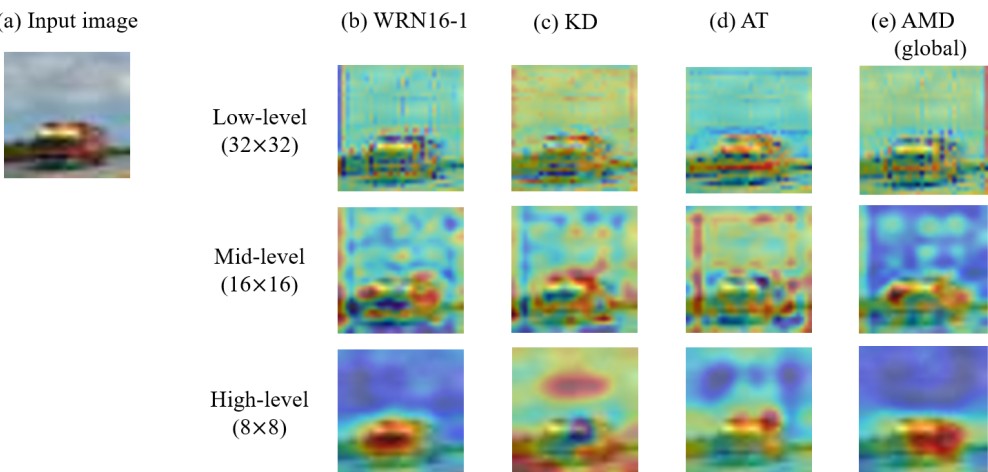

Figure 9: Activation maps for different levels of students (WRN16-1) trained with a teacher (WRN16-3) on CIFAR-10.

### 5.5.1 Activation maps for the different levels of layers

The activation maps from intermediate layers with various methods are shown in Figure 9. The proposed method, AMD, shows intuitively similar activated regions to the traditional KD (Hinton et al., 2015) in the low-level. However, at mid-level and high-level, the proposed method represents the higher activations around the region of a target object, which is different from the previous methods (Hinton et al., 2015; Zagoruyko & Komodakis, 2017). Thus, the proposed method can classify positive and negative areas more discriminatively, compared to the previous methods (Hinton et al., 2015; Zagoruyko & Komodakis, 2017). The high-level activation maps with various input images are described in Figure 10. The activation from proposed method is seen to be more centered on the target. The result shows that the proposed method performs better in focusing on the foreground object distinctly with high weight, while being less distracted by the backgroun compared to other methods (Hinton et al., 2015; Zagoruyko & Komodakis, 2017). With higher weight over regions of interest, the student from the proposed method has a stronger discrimination ability. Therefore, the proposed method guides student models to increase class separability.

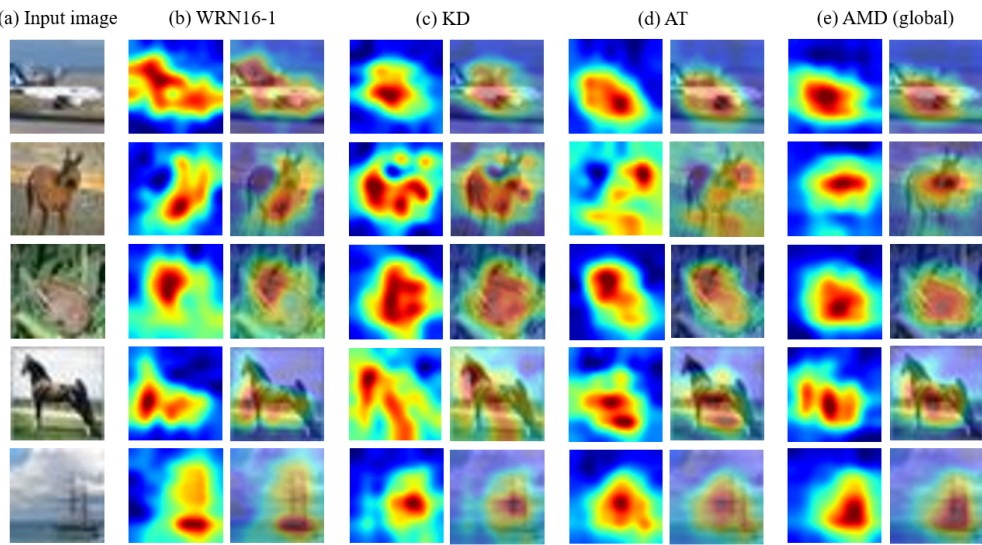

Figure 10: Activation maps of high-level from students (WRN16-1) trained with a teacher (WRN16-3) for different input images on CIFAR-10.

### 5.5.2 Activation maps for global and local distillation of AMD

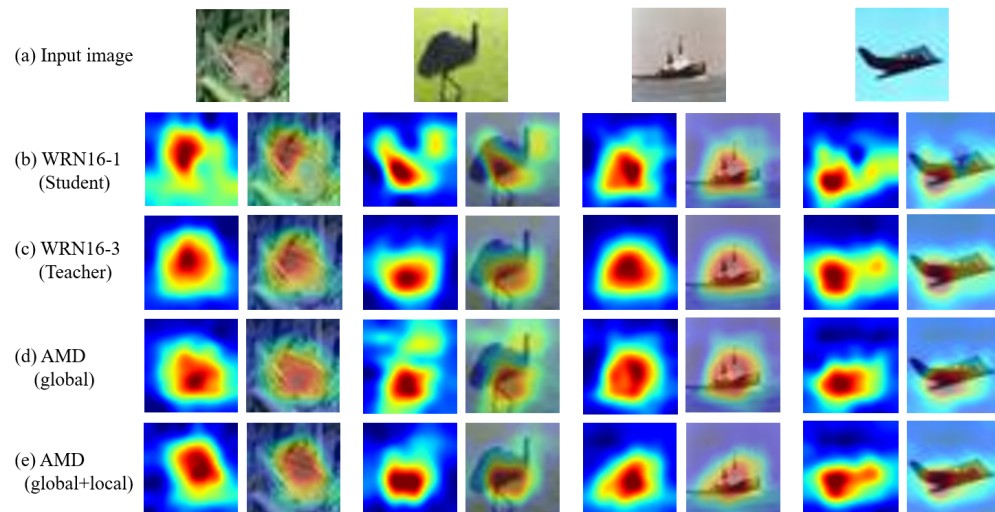

Figure 11: Activation maps of high-level from students (WRN16-1) for AMD trained with a teacher (WRN16-3) for different input images on CIFAR-10.

To investigate the impact of using global and local features for AMD, we illustrate relevant results in Figure 11. When both global and local features are used for distillation, the activated area is located and shaped more similar to the teacher, than using the global feature only. Also, AMD (global+local) focuses more on the foreground object with higher weights than AMD (global). AMD (global+local) guides the student to focus more on the target regions and finds discriminative regions. Thus, using global and local features is better than using global features alone for the proposed method.

## 5.6 Combining with Existing Methods

Even if a model shows good performance in classification, it may have miscalibration problems (Guo et al., 2017) and does not always obtain improved results from combining with other robust methods. In this section, to evaluate generalizability of models trained by each method and to explore if the method can complement other methods, we implement experiments with various existing methods. We use the method in various ways to demonstrate how easily it can be combined with any previous learning tasks. We trained students with fine-grained features (Wang et al., 2019; 2020a), Mixup (Zhang et al., 2018) augmentation, and one of the baselines such as SP (Tung & Mori, 2019) that is not based on the attention feature based KD. WRN16-1 students were trained with WRN16-3 and WRN28-1 teachers. We examine whether the proposed method can be combined with other techniques and compare the results to baselines.

### 5.6.1 Fine-grained feature-based distillation

If the features of teacher and student are compatible, it results in a student achieving 'minor gains' (Wang et al., 2019). To perform better distillation and to overcome the problem of learning minor gains, a technique for generating a fine-grained feature has been used (Wang et al., 2019; 2020a). For distillation with AMD and creating the fine-grained (masked) feature, a binary mask is adopted when the negative feature is created. For example, if the probability of the point for the negative map is higher than 0.5, the point is multiplied by 1, otherwise by 0. Then, compared to non-masking, it boosts the difference between teacher and student, where the difference can be more focused on loss function for training. The results for AMD with or without using masked feature-based distillation are presented in Figure 12. The parameter $\gamma$ for training a student based on AMD without masked features is 5000 for all setups across datasets. When masked features are used for AMD, to generate the best results, $\gamma$ of 3000 is applied to setup (b) on CIFAR-10, setup ($c^a$) on CINIC-10, and all setups on Tiny-ImageNet. For CIFAR-10, AMD (global+local) without masked features has the best performing result in most cases. AMD (global+local) with masked features shows the best with setup (d). For CINIC-10, the results of AMD with masked features for setup (d) show the best. For Tiny-

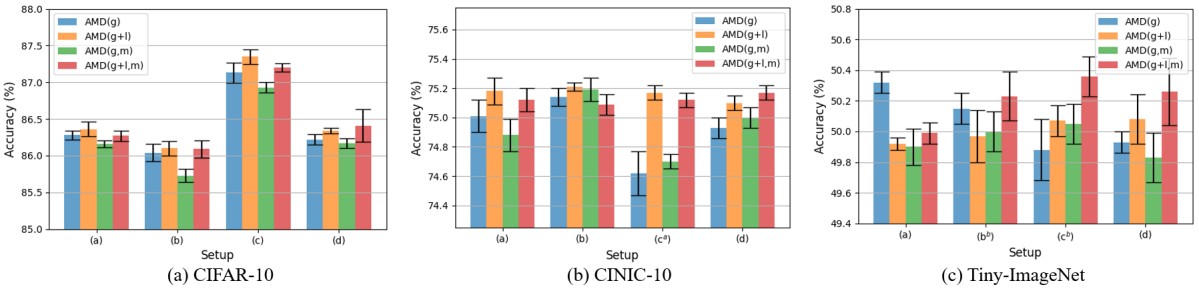

Figure 12: Accuracy (%) from students (WRN16-1) for AMD trained with a teacher (WRN16-3) with/without masked features. "g", "l", and "m" denote global, local, and masked feature, respectively.

ImageNet, in most cases, AMD with masked features performs the best. Therefore, when the complexity of a dataset is high, fine-grained features can help more effectively improve the performance, and the smaller parameter of $\gamma$, 3000, generates better accuracy. Also, AMD (global+local) with masked features produces better performance than AMD (global) with the one. For setup (d) – different architectures for teacher and student – with/without masked features, AMD (global+local) outperforms AMD (global). This could be due to the fact that the teacher's features differ from the student's because the two networks have different architectures, resulting in different distributions. So, masked features with both global and local distillation influence more on setup (d) than other setups. The difference between AMD (global) and AMD (global+local) with masked features is also discriminatively shown with the harder problem in classification. If the student's and teacher's architectural styles are similar, the student is more likely to achieve plausible results (Wang & Yoon, 2021).

### 5.6.2 Applying augmentation methods

Mixup (Zhang et al., 2018) is one most commonly used augmentation methods. We demonstrate here that AMD complements Mixup. Mixup's parameter is set to $\alpha_{\text{Mixup}} = 0.2$. A teacher is trained with the original training set and learns from scratch. A student is trained with Mixup and the teacher model is implemented as a pre-trained model.

As described in Figure 13, with Mixup, most of the methods generate better results. However, when a WRN28-1 teacher is used, the performance of the student from AFD is degraded. Also, compared to the baselines, AMD obtains more gains from Mixup. To study the generalizability and regularization effects of Mixup, we measured expected calibration error (ECE) (Naeini et al., 2015; Guo et al., 2017) and negative log likelihood (NLL) (Guo et al., 2017) for each method. ECE is a metric to measure calibration, representing the reliability of the model (Guo et al., 2017). A probabilistic model's quality can be measured by using NLL (Guo et al., 2017). The results of training from scratch with Mixup show a higher ECE and NLL than the results of training without Mixup, as seen in Table 8. However, the methods, including knowledge distillation, generate lower ECE and NLL. This implies that knowledge distillation from teacher to student influences the generation of a better model not only for accuracy but also for reliability. In both (a) and (b), with Mixup, AMD (global+local) shows robust calibration performance. Therefore, we confirm that an augmentation method such as Mixup gets the benefits from AMD in generating better calibrated performance. As can be seen in Figure 14, WRN16-1 trained from scratch with Mixup produces underconfident predictions (Zhang et al., 2018), compared to KD (Hinton et al., 2015) with Mixup. AMD (global+local) with Mixup achieves the best calibration performance. These results support the advantage of AMD, that it can be easily combined with common augmentation methods to improve the performance in classification with good calibration.

### 5.6.3 Combination with other distillation methods

To demonstrate how AMD can perform with the other distillation methods, we adopt SP (Tung & Mori, 2019) which is not an attention based distillation method. A teacher is trained with the original training set and learns from scratch. SP (Tung & Mori, 2019) is applied while a student is being trained. We compare with baselines, depicted in Figure 15. In all cases, with SP, the accuracy is increased. Compared to the other attention based methods, AMD gets more gains by SP. Therefore, AMD can be enhanced and can perform well with the other distillation methods such as SP. We additionally analyzed the reliability described in Table 9. AMD (global+local) with SP shows the lowest ECE

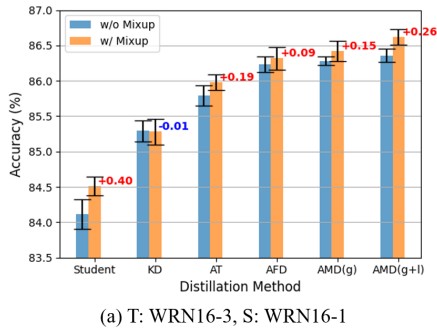 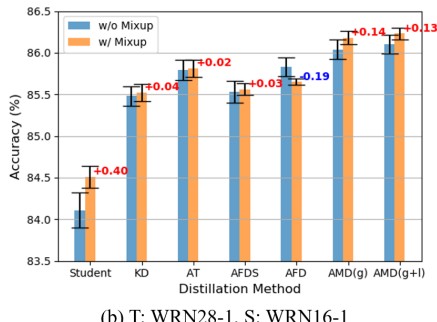

(a) T: WRN16-3, S: WRN16-1        (b) T: WRN28-1, S: WRN16-1

Figure 13: Accuracy (%) of students (WRN16-1) for knowledge distillation methods, trained with Mixup and a teacher (WRN16-3) on CIFAR-10. "T" and "S" denote teacher and student, respectively. "g" and "l" denote using global and local feature distillation, respectively. "Student" is a result of WRN16-1 trained from scratch.

Table 8: ECE (%) and NLL (%) for various knowledge distillation methods with Mixup on CIFAR-10. "g" and "l" denote using global and local feature distillation, respectively. The results (ECE, NLL) for WRN16-3 and WRN28-1 teachers are (1.469%, 44.42%) and (2.108%, 64.38%), respectively.

| Setup | Method | w/o Mixup | | w/ Mixup | |
|-------|--------|-----------|-----|----------|-----|
| | | ECE | NLL | ECE | NLL |
| | Student | 2.273 | 70.49 | 7.374 (+5.101) | 90.58 (+20.09) |
| (a) | KD (Hinton et al., 2015) | 2.065 | 63.34 | 1.818 (-0.247) | 55.62 (-7.71) |
| | AT (Zagoruyko & Komodakis, 2017) | 1.978 | 60.48 | 1.652 (-0.326) | 50.84 (-9.64) |
| | AFD (Ji et al., 2021) | **1.890** | **56.71** | 1.651 (-0.240) | 50.22 (-6.49) |
| | AMD (g) | 1.933 | 59.67 | 1.645 (-0.288) | 50.33 (-9.34) |
| | AMD (g+l) | 1.895 | 57.60 | **1.592** (-0.304) | **49.68** (-7.92) |
| (b) | KD (Hinton et al., 2015) | 2.201 | 68.75 | 1.953 (-0.249) | 58.81 (-9.93) |
| | AT (Zagoruyko & Komodakis, 2017) | 2.156 | 67.14 | 1.895 (-0.261) | 56.51 (-10.62) |
| | AFDS (Wang et al., 2020b) | 2.197 | 68.53 | 1.978 (-0.219) | 58.86 (-9.68) |
| | AFD (Ji et al., 2021) | 2.143 | 66.05 | 1.900 (-0.243) | 57.68 (-8.37) |
| | AMD (g) | **2.117** | **66.47** | 1.869 (-0.248) | 56.05 (-10.42) |
| | AMD (g+l) | 2.123 | 67.51 | **1.853** (-0.270) | **55.15** (-12.36) |

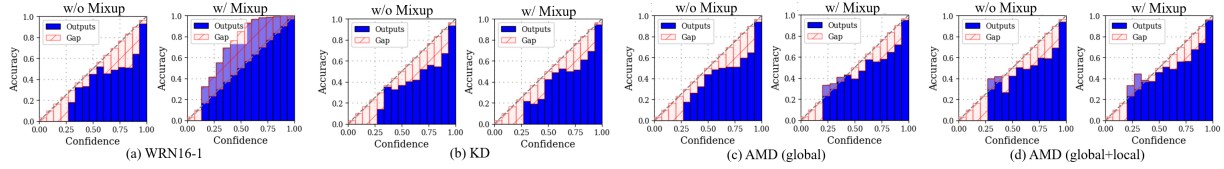

(a) WRN16-1       (b) KD       (c) AMD (global)       (d) AMD (global+local)

Figure 14: Reliability diagrams of students (WRN16-1) for knowledge distillation methods, trained with Mixup and a teacher (WRN16-3) on CIFAR-10. For the results of each method, the left is the result without Mixup, and the right is with Mixup.

and NLL values. It verifies that AMD with SP can generate a model having higher reliability with better accuracy. Thus, the proposed method can be used with an additional distillation method. Also, the proposed method with SP can perform with different combinations of teacher and student with well-calibrated results. As illustrated in Figure 16, with SP (Tung & Mori, 2019), AT (Zagoruyko & Komodakis, 2017) and AFD (Ji et al., 2021) produce more overconfident predictions, compared to AMD (global+local) with SP (Tung & Mori, 2019) that gives the best calibration performance. Conclusively, our empirical findings reveal that AMD can perform with other distillation methods such as SP (Tung & Mori, 2019) to generate more informative features for distillation from teacher to student.

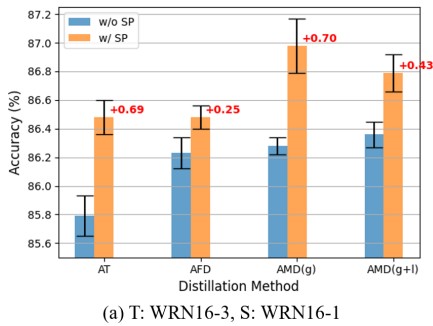
(a) T: WRN16-3, S: WRN16-1

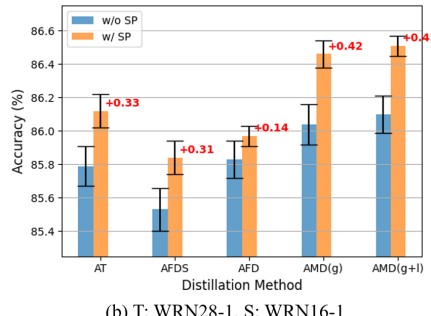
(b) T: WRN28-1, S: WRN16-1

Figure 15: Accuracy (%) of students (WRN16-1) for knowledge distillation methods, trained with SP and a teacher (WRN16-3) on CIFAR-10. "T" and "S" denote teacher and student, respectively. "g" and "l" denote using global and local feature distillation, respectively. "Student" is a result of WRN16-1 trained from scratch.

Table 9: ECE (%) and NLL (%) for various knowledge distillation methods with SP on CIFAR-10. "g" and "l" denote using global and local feature distillation, respectively. The results (ECE, NLL) for WRN16-3 and WRN28-1 teachers are (1.469%, 44.42%) and (2.108%, 64.38%), respectively.

| Setup | Method | w/o SP | | w/ SP | |
|---|---|---|---|---|---|
| | | ECE | NLL | ECE | NLL |
| (a) | AT (Zagoruyko & Komodakis, 2017) | 1.978 | 60.48 | 1.861 (-0.118) | 56.22 (-4.26) |
| | AFD (Ji et al., 2021) | **1.890** | **56.71** | 1.881 (-0.010) | 56.73 (-0.02) |
| | AMD (g) | 1.933 | 59.67 | 1.808 (-0.125) | 54.74 (-4.93) |
| | AMD (g+l) | 1.895 | 57.60 | **1.803** (-0.092) | **53.80** (-3.80) |
| (b) | AT (Zagoruyko & Komodakis, 2017) | 2.156 | 67.14 | 2.095 (-0.060) | 65.38 (-1.75) |
| | AFDS (Wang et al., 2020b) | 2.197 | 68.53 | 2.128 (-0.069) | 66.61 (-1.92) |
| | AFD (Ji et al., 2021) | 2.143 | 66.05 | 2.118 (-0.024) | 65.39 (-0.66) |
| | AMD (g) | **2.117** | **66.47** | 2.058 (-0.059) | 63.37 (-3.10) |
| | AMD (g+l) | 2.123 | 67.51 | **2.043** (-0.080) | **63.23** (-4.28) |

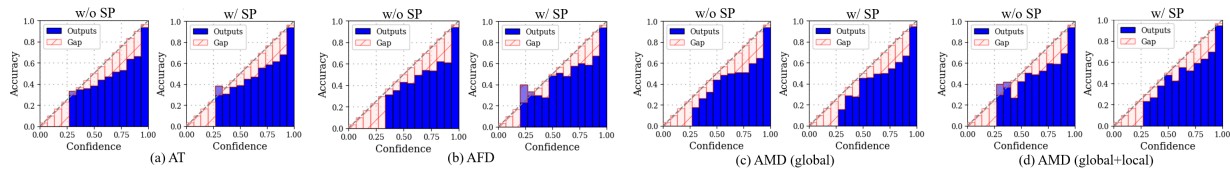
(a) AT      (b) AFD      (c) AMD (global)      (d) AMD (global+local)

Figure 16: Reliability diagrams of students (WRN16-1) for knowledge distillation methods, trained with SP and a teacher (WRN16-3) on CIFAR-10. For the results of each method, the left is the result without SP, and the right is with SP.

## 6 Conclusion

In this paper, we proposed an angular margin based knowledge distillation (AMD) method. Our analysis shows that the proposed method trains the student model effectively in KD. Through multiple combinations of the models and showing great performance on the more challenging dataset, we have verified the robustness of the proposed approaches. We have presented the effects of using global and local feature distillation for AMD with activation maps. Furthermore, we have confirmed that the proposed method can be easily combined with previous studies. Fine-grained features can be applied to AMD to obtain better performance. Also, other approaches, such as Mixup and SP, can be implemented with AMD for better performance and lower calibration error.

In future work, we aim to extend the idea presented in this paper to explore the distillation effects with different hypersphere feature embedding methods (Wang et al., 2018b; Deng et al., 2019). In addition, our approach could provide insights for further advancement in other applications such as object detection and semantic segmentation.

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
