# OpenReview forum: "AMD: Angular Margin based Knowledge Distillation"
_TMLR — Rejected by TMLR_

### Review · Reviewer_H6jA · 2022-04-16

**Summary Of Contributions:**

This paper proposes a new approach for knowledge distillation for image classification. The method can be broadly characterized as an attentioned transfer approach (Zagoruyko & Komodakis, 2017), i.e., a teacher network transfers its knowledge to the students through transferring of attention maps. Zagoruyko & Komodakis 2017 proposed to match (minimize the norm difference) between teacher and student’s normalized attention maps. This paper presents an alternative way of matching the attention maps. Inspired by angular margin based loss (Choi et al. 2020), this paper proposes to match the angular distance between “positive/negative features”. Furthermore, they also propose to split each feature map into four local parts and found it to improve performance.

The paper empirically validates their proposed method on CIFAR10, CINIC-10 and Tiny-ImageNe and found it to outperform recent baselines. They also presented sensitivity analysis on the hyperparameters and provided visualization results to illustrate the between matching of attention maps. Finally, they demonstrate that the proposed approach can be combined with other existing methods to further improve the model performance.


**Broader Impact Concerns:**

No concerns on the ethical implications. Model distillation and image classification are standard applications of machine learning and have been widely studied.

**Requested Changes:**

My major concern is with the description of the approach. Currently, the paper does not adequately describe their approach and a reader would not be able to reproduce their work. Specifically, the math notations aren’t precise and the motivation of the proposed method is lacking. Next, the paper’s result can be strengthened by conducting experiments on datasets that can be directly compared with prior works. It is unclear how meaningful/interesting are the current presented results.

Misc.
- Typo in 3.4 “Angular Marin based Distillation”.
- Add color bars to Fig. 7, 8, and 9.
- Page. 7
> "We determine the parameters empirically, considering the distillation effects by the capacity of models. A more detailed description of parameters appears in section 4.4."

   For that section, the "parameters" should be hyper-parameters to be consistent with Sec. 4.4.


**Strengths And Weaknesses:**

# A. Strengths
### A1. This paper motivates the problem of knowledge distillation well. Knowledge distillation is relevant to the community and has practical applications to the real-world, e.g., making models light-weight, and more efficient.

### A2. The paper clearly stated their contribution and conducted a variety of experiments to validate their performance gain.

### A3. The related work section sufficiently reviewed the related works and discussed their relevance.

# B. Weaknesses
### B1. The approach section (Sec. 3) requires significant rewriting.
(a) Sec. 3.2, $f_T^l = \sum_{j=1}^c |A_{T,j}^l|^2_T$, the subscript $T$ on the absolute value is confusing. What does it mean?
(b) Sec. 3.3 is very imprecise. Personally, I do not think a reader can adequately reproduce the work based on the description. Just to give a few examples:
> “between the positive (relevant to the target object) and negative features (irrelevant)

How is “relevant” defined and how to extract these features?

> “First, for a given input feature vector x, the output activation of the layer with weight $W$ is given by ....for the positive map”

This entire paragraph is confusing. What is the “input feature vector x” in this case? What is its dimension?
"The layer with weight $W$", is this the weight of a convolution layer? Is it a matrix or a vector?
What does “for the positive map” mean? It's not defined anywhere? Please define these terms properly.

> We begin by reformulating positive and negative features... Eq.(4)

Eq. (4) is also confuses me. On the right hand side, there is a subscript $i$ but not the left hand side. Is $\tilde{G}_{ang1}$ a vector or a matrix or a tensor?
Eq. 7, has the Frobenius and 2-norm, but it remains unclear to me if $G$ is a matrix or a vector? Additionally, Eq. 9 leads to further confusion, when feature maps are concatenated. In general, please properly define the introduced notation, otherwise the paper is challenging to understand.

 ### B2. Motivation of the proposed Angular Margin Metric is unclear.
The formulation in Eq. (5) and Eq. (6) is not very well motivated. Why is the margin penalty term multiplied inside the cosine? Also the text says “by adding an angular margin penalty”, but Eq. 6 is a multiplication.


### B3. Experiments not evaluated on the same datasets as prior works.
Prior works also experimented with CIFAR100 and ImageNet dataset. The results would be more helpful if they can be directly compared with the reported numbers from prior works. Additionally, evaluation on ImageNet would be an interesting result to have.

### B4. The paper claims that the proposed method leads to better attention maps and presents a few visualizations.

Is it possible to get a quantitative metric to directly evaluate this?

### B5. Code is not included with the supplemental materials.
The description of the experiment and approach may be difficult to reproduce. Will the author release their code? This would help with the reproducibility.

### B6. Organization of the experiments
The inclusion of hyper-parameters description in Sec. 4.1 (Dataset Description) seems very strange to me. Also, please also describe the hyper-parameter search procedure, how it was searched and over what range.
Maybe organize the experiment section based on the dataset? Currently, the reader has to go through a lot of experiment details for all the datasets. It might be helpful to introduce the setup and represent the result.

---

> ### Author Response · Authors · 2022-05-18
> **Response to Reviewer H6jA**
>
> We thank the reviewer for their helpful review and feedback on how to improve our work. We have responded to specific comments below.
>
> **B1.** The approach section (Sec. 3) requires significant rewriting.
>
> * We rewrote the section of the proposed method to explain our approach and novelty. Based on the comments, we updated the relevant parts of the equation.
>
>
> **B2.**  Motivation of the proposed method
>
> * We rewrote the section of the proposed method to explain our approach and novelty. Based on the comments, we updated equation parts and added a figure (Figure 2) to explain our method and previous method (AT).
>
>
> **B3.** Experiments not evaluated on the same datasets as prior works.
>
> * We added the results on ImageNet-1k in Table 6. AMD (global) shows better results than other methods we explored, increasing the top-1 and top-5 accuracy by 1.83% and 1.43% over the results of learning from scratch, respectively.
>
>
> **B4.** The paper claims that the proposed method leads to better attention maps and presents a few visualizations.
>
> * Unfortunately, we are not aware of a quantitative evaluation for this visualization. Since our proposed method aims to mimic the teacher’s behavior based on the teacher’s attention map, through this visualization, we wanted to show qualitatively the performance of our method. We follow the same way as the previous study (baseline method, AT [1]).
>
>
> **B5.** Code is not included with the supplemental materials.
> * We will release our code after paper acceptance.
>
>
> **B6.** Organization of the experiments
> * Thanks for your suggestion. We modified the section of data description and experimental settings. We carefully explain the hyperparameter and the settings of training/testing procedure in Section 5.1.2. To reproduce the baseline results, we follow the same hyper-parameter settings as suggested in previous works  [2, 3, 4]. For our method, we conduct sensitivity analysis for our hyperparameters in section 5.4 and choose the hyperparameters that achieve the best accuracy.
>
> [1] Sergey Zagoruyko and Nikos Komodakis. Paying more attention to attention: Improving the performance of convolutional neural networks via attention transfer. In Proceedings of the International Conference on Learning and Representations, pp. 1–13, 2017. \
> [2] Jang Hyun Cho and Bharath Hariharan. On the efficacy of knowledge distillation. In Proceedings of the IEEE/CVF International Conference on Computer Vision, pp. 4794–4802, 2019. \
> [3] Frederick Tung and Greg Mori. Similarity-preserving knowledge distillation. In Proceedings of the IEEE/CVF International Conference on Computer Vision, pp. 1365–1374, 2019. \
> [4] Mingi Ji, Byeongho Heo, and Sungrae Park. Show, attend and distill: Knowledge distillation via attention-based feature matching. In Proceedings of the AAAI Conference on Artificial Intelligence, volume 35, pp. 7945–7952, 2021.

---

### Review · Reviewer_LH2Q · 2022-04-24

**Summary Of Contributions:**

This paper proposes to involve additional angular distance losses on the intermediate representations of networks to obtain attentive features for knowledge distillation. The whole objective consists of angular margin loss, cross-entropy loss, and knowledge distillation loss for training the student model. Experiments are conducted on three datasets: CIFAR-10, CINIC-10 and Tiny-ImageNet, to demonstrate the effectiveness of the proposed distillation method.

**Broader Impact Concerns:**

No.

**Requested Changes:**

It will be appreciated if  the authors can revise their submission on the following aspects:

*1) Provide more detailed results on ImageNet-1K under the fair experimental setting. This is the most important factor for me to give the final recommendation.*

 *2) Explain how to determine the coefficients of different AM losses on different layers.*

 *3) Provide more ablation results with or without the early stop technique.*

  *4) Give more explanations from the mechanism or theory perspective why the angular distance will be better than other distance metrics, such as $\ell_2$, KL, etc. for distilling features.*

**Strengths And Weaknesses:**

### Strengths:

The proposed distillation framework is simple, clear, and easy to follow.

### Weaknesses:

The idea of using the angular distance on the intermediate features is fairly simple, I think this can be a merit for the paper if it is well supported by the experiments, while I’m a little bit disappointed as the paper only performs experiments on the small datasets like CIFAR and Tiny-ImageNet, without regular ImageNet-1K, also the improvement on these small datasets seems marginal and not significant.

    1) My main concern of this paper is the experimental results, which are not sufficient and also not significant to verify the effectiveness of the proposed method. Considering that the proposed method is extremely simple, I think ImageNet-1K results are necessary to support the statements in this paper.

    2) I’m also a little bit confused about the procedure of the distillation framework. In section 3, this paper mentioned “1. Training teacher and student models based on early stopped KD (Cho & Hariharan, 2019).” It seems many compared methods (including the vanilla KD) in the experimental section did not involve this early stop trick, so it is questionable whether the comparisons in this paper are fair, maybe the slight improvement of the proposed method on the small datasets is just from using this additional early stop technique.

    3) On CIFAR-10 (in Table 3), the improvement seems marginal and basically in the variance from different runs. Frankly, the results are not solid and convincing to support that the proposed angular distance is effective for distillation and can bring better performance.

    4) It is not clear how to determine the coefficients of different AM losses on different layers. From my experience on knowledge distillation experiments, the intermediate features have a limited effect on the performance than the final logits distillation.

**Overall,** the submission can be an interesting work if the authors can provide convincing results on the larger datasets like ImageNet-1K. Otherwise, I think the value of this paper for the relevant area will be really limited, and basically, I just feel this paper is not ready for publication with the current shape.

---

> ### Author Response · Authors · 2022-05-18
> **Response to Reviewer LH2Q**
>
> We thank the reviewer for their helpful review and feedback. We have responded to comments below.
>
> **1)** Experimental results on ImageNet-1k
>
> * We have now added results on ImageNet-1k in Table 6. AMD (global) shows better results than other methods we explored, increasing the top-1 and top-5 accuracy by 1.83% and 1.43% over the results of learning from scratch, respectively. The experimental setting is explained in section 5.1.2.
>
> **2)** Comparisons of early stopped KD and vanilla KD
>
> * To address this issue, we have now provided several results from ESKD and Full KD testing to explain why we chose ESKD methods for our experiments. We add Figure 4 which is to compare ESKD with KD. As shown in Figure 4, training with ESKD tends to show better performance in most cases. Regardless of whether early stopped KD is used, AMD always outperformed KD and AT.
>
> **3)** On CIFAR-10 (in Table 3), the improvement is marginal
>
> * For the experiments, all baselines and ours utilize the standard KD method. To present additional results, we have added experimental results on ImageNet-1k in Table 6. In ImageNet experiments, the proposed method improved the performance by 1.83% compared to the model trained from scratch. We do however note that, the difference between accuracies of methods could be small, but we ran sufficiently many times and reported the averaged accuracy. Also, marginal difference is commonly reported in prior works [1, 2], even with improvement sometimes less than 0.1%, since distillation is more directly about model compression.
>
> **4)** Coefficients of different AM losses on different layers
>
> * In the proposed method section, we explained how we utilize features from different layers. AM losses from different layers are summed and divided by the total number of pairs in the layers.
>
> **5)** More explanations from the mechanism or theory perspective why the angular distance will be better
>
> * Many previous works [3, 4, 5] have shown that angular metric has the benefit of learning better discriminative feature representation in networks. We were motivated by previous works and applied it to this new type of knowledge for KD. Furthermore, as described in Figure 5, we conducted experiments for each component (A, P, N in Equation (7)) of the proposed AMD loss and observed that the component (A) induced the angular distance shows the strongest influence on the performance of KD (i.e. A>P>N).  Finally, the combination of all components yielded the highest performance.
>
>
> [1] Sergey Zagoruyko and Nikos Komodakis. Paying more attention to attention: Improving the performance of convolutional neural networks via attention transfer. In Proceedings of the International Conference on Learning and Representations, pp. 1–13, 2017. \
> [2] Jang Hyun Cho and Bharath Hariharan. On the efficacy of knowledge distillation. In Proceedings of the IEEE/CVF International Conference on Computer Vision, pp. 4794–4802, 2019. \
> [3] Weiyang Liu, Yandong Wen, Zhiding Yu, Ming Li, Bhiksha Raj, and Le Song. Sphereface: Deep hypersphere embedding for face recognition. In Proceedings of the IEEE Conference on Computer Vision and Pattern Recognition, pp. 212–220, 2017. \
> [4] Weiyang Liu, Yandong Wen, Zhiding Yu, and Meng Yang. Large-margin softmax loss for convolutional neural networks. In Proceedings of the International Conference on Machine Learning, volume 48, pp. 507–516, 2016 \
> [5] Feng Wang, Jian Cheng, Weiyang Liu, and Haijun Liu. Additive margin softmax for Face verification. IEEE Signal Processing Letters, 25(7):926–930, 2018a.

---

### Review · Reviewer_ALGP · 2022-05-02

**Summary Of Contributions:**

This paper works on knowledge distillation (KD) for classification problems, aiming to transfer the knowledge from a stronger teacher model (e.g., a model with more capacity) to the student model. The authors propose a new distillation loss term (angular margin-based distillation), which is to transfer or enforce the angular margin between the positive and negative features maps. The authors conducted experiments and analyses on three “small-scale” datasets (CIFAR-10, CINIC-10, and Tiny-ImageNet) to demonstrate the effectiveness of the proposed approach.

**Broader Impact Concerns:**

No concerns. Knowledge distillation is widely studied in the field.

**Requested Changes:**

Please see the above weakness. I have provided detailed comments and several suggestions for how the authors can improve the paper. Specifically, the paper lacks clear contributions, novelty, and insights. Section 3 needs improvement in writing and details. Section 4 needs more baseline, datasets, and explanations of the conducted studies and improvements. Based on the response, I could then have a better understanding of the novelty and contribution of the paper.

Some typos, for example, P4: “leveraging to its effect”


**Strengths And Weaknesses:**

## Strengths

S1. The paper is in general well-written (in terms of word usage and grammar error).


S2. The paper conducts extensive analyses and ablation studies.


## Weaknesses

W1. The contribution, novelty, and significance of the paper are limited or unclear at this moment.

-	It is not unclear about the technical novelty as it seems to combine the existing methods of knowledge distillation in feature maps and the angular similarity/dissimilarity. Angular similarity (e.g., cosine distance) has been widely studied in many CV and ML problems and is more or less a hyperparameter (e.g., Euclidean distance or cosine distance) in implementing any distance-based loss. The positive and negative maps seem interesting, but there are no further details on how to implement them. Overall, the paper lacks additional insights on applying the angular-based loss to knowledge distillation.

-	Figure 1 is not informative as it cannot really tell the novelty and/or details of the proposed method. The caption reads like a normal feature-based distillation method is doing.

-	I’m not fully convinced about the “dual” ability part of the argument. A teacher model also uses the feature/attention map to differentiate positive and negative examples/regions. If so, why can’t the student learn such an ability by matching the teacher’s feature map?

-	Please see more details in the following comment.


W2. The technical part of the paper needs significant improvement. It is very hard to understand the novelty, contribution, and implementation details in the current version.

-	Section 3.2 is very unclear, yet it is the basis of the proposed method. Specifically, is “$f_T^L$” computed for each spatial location or averaged across spatial locations (i.e., h and w)? What is the loss used to compare students and teachers? Obviously, it should not be KL divergence, but the authors did not provide details. Also, the subscript of T and S after “|” is a bit annoying. The subscript after “|” or “||” is usually used to denote the norm type.

-	Throughout section 3.3, I have little idea of how to construct the “positive” and “negative” maps. Do the authors need to re-train the teacher model specifically to provide these or not?

-	I also do not clearly see the motivations why the authors want to define and match Eq. (4)-(6), and why adding “m”? Does “m” have a similar effect to the temperature “\tau”?

-	For sections 3.4 and 3.5, since I am not sure if the loss introduced in section 3.2 is computed per pixel (or feature grid location h and w) or not, it is hard for me to grasp the idea of local distillation. I would suggest that the authors provide a figure for better understanding, like the ones in https://arxiv.org/pdf/2103.14030.pdf (Fig. 1). BTW, what is the “K” used in the experiments?

-	Overall, section 3 needs significant improvement. As a journal submission, the authors should provide more details such that the readers can reproduce the results. I would also suggest that the authors list the objective functions of highly relevant baseline methods.

W3. While the authors compared several baselines and performed several studies in the experiment section (i.e., section 4), it is hard to understand why the proposed method performs better or the purpose of the studies.

-	Since all the compared methods more or less follow Eq (1), I would suggest that the authors clearly list down all the compared methods’ objective functions to provide a detailed comparison.

-	What is the purpose of section 4.6.3? It went across 2~3 pages (P 14-16) but I can hardly understand its purpose or importance.

W4. Experiments are not convincing.

-	The authors only conducted experiments on small-scale datasets. Even the paper by Sergey Zagoruyko and Nikos Komodakis in 2017 has experimented on ImageNet, CUB, and Scenes. The authors should consider more datasets in terms of their scale and diversity to justify the effectiveness of the proposed method.

-	The teacher networks seem not very strong (e.g., ResNet44). I would suggest that the authors consider stronger teacher models, like the ones studied in https://arxiv.org/pdf/2006.10029.pdf

-	The improvement is quite small. In many of the settings, the improvement over the baselines is < 1.

-	If possible, more baseline methods should be compared. For now, only AFDS and AFD are proposed after 2020. Given that knowledge distillation is a widely studied subfield in ML, I would expect that more recent baselines are involved. Or, at least the authors should provide a reason why more recent methods are not compared.

W5. The related work is well-written but can contain some more works. For example, several compared baselines (e.g., Yim et al.) are not discussed in the related work. Besides, I would like the authors to clearly position their work in the context of the related work. . For example, what is the difference between the proposed method and related work?

---

> ### Author Response · Authors · 2022-05-18
> **Response to Reviewer ALGP [Part1]**
>
> We thank the reviewer for their helpful review and suggestion on how to improve our work. We have responded to comments below.
>
> **W1**
>
> **Q.** It is not unclear about the technical novelty as it seems to combine the existing methods of knowledge distillation in feature maps and the angular similarity/dissimilarity. Angular similarity (e.g., cosine distance) has been widely studied in many CV and ML problems and is more or less a hyperparameter (e.g., Euclidean distance or cosine distance) in implementing any distance-based loss. The positive and negative maps seem interesting, but there are no further details on how to implement them. Overall, the paper lacks additional insights on applying the angular-based loss to knowledge distillation.
>
> **A.** We rewrote the proposed method part with a newer formulation. Also, we added a section of Background and a modified section on “The Proposed Method” to explain our method more clearly. In Figure 5, we tested with each component of the loss from equation (7) and explained angular probability’s contribution to the performance. Among three components of the AM loss function, A achieves the best (i.e. A>P>N). When all components are used for distillation (AMD), the best accuracy is achieved.
> For more details for angular-based loss, we explained in the section on Related work and Background, including references using angular probability [1, 2, 3]. Further, we have provided a new figure (Figure 2), and modified section 4 to explain how we utilize the angular margin for distillation for details.
>
> [1] Weiyang Liu, Yandong Wen, Zhiding Yu, Ming Li, Bhiksha Raj, and Le Song. Sphereface: Deep hypersphere embedding for face recognition. In Proceedings of the IEEE Conference on Computer Vision and Pattern Recognition, pp. 212–220, 2017. \
> [2] Weiyang Liu, Yandong Wen, Zhiding Yu, and Meng Yang. Large-margin softmax loss for convolutional neural networks. In Proceedings of the International Conference on Machine Learning, volume 48, pp. 507–516, 2016. \
> [3] Feng Wang, Jian Cheng, Weiyang Liu, and Haijun Liu. Additive margin softmax for Face verification. IEEE Signal Processing Letters, 25(7):926–930, 2018a.
>
> **Q.** Figure 1 is not informative as it cannot really tell the novelty and/or details of the proposed method. The caption reads like a normal feature-based distillation method is doing.
>
> **A.** To explain the procedure in a more informative manner, we have added a new figure (Figure 2) to the proposed method description in section 4.
>
> **Q.** I’m not fully convinced about the “dual” ability part of the argument. A teacher model also uses the feature/attention map to differentiate positive and negative examples/regions. If so, why can’t the student learn such an ability by matching the teacher’s feature map?
>
> **A.** To provide the detailed effects of our proposed method, we have added simple testing results for explaining AMD loss in Figure 5. As shown in the figure, all components affect the result. And, when all components are included (AMD), it shows the best.
>
> ***
> **W2**
>
> **Q.** Section 3.2 is very unclear, yet it is the basis of the proposed method. Specifically, is “$f_T^L$” computed for each spatial location or averaged across spatial locations (i.e., h and w)? What is the loss used to compare students and teachers? Obviously, it should not be KL divergence, but the authors did not provide details. Also, the subscript of T and S after “|” is a bit annoying. The subscript after “|” or “||” is usually used to denote the norm type.
>
> **A.** Thank you for the comment. We have reorganized the section on Related Work and Background and have also edited the relevant details mentioned in this comment.
>
> **Q.** Throughout section 3.3, I have little idea of how to construct the “positive” and “negative” maps. Do the authors need to re-train the teacher model specifically to provide these or not?
>
> **A.** We did not retrain the teacher model. Positive and negative feature maps are simply obtained from the selected layer’s outputs. To make it clearer, we added a figure and explanation in the section of Proposed Method.
>
> **Q.** I also do not clearly see the motivations why the authors want to define and match Eq. (4)-(6), and why adding “m”? Does “m” have a similar effect to the temperature “\tau”?
>
> **A.** We have modified the equations referred to above. As explained in the introduction and proposed method “m” is a constant (hyper-parameter) for inducing a margin to angle. It creates a gap between positive and negative features that makes positive features more attentive. We present the effects of “m” on the accuracy in the section titled Sensitivity analysis.

---

> ### Author Response · Authors · 2022-05-18
> **Response to Reviewer ALGP [Part2]**
>
> **Q.** For sections 3.4 and 3.5, since I am not sure if the loss introduced in section 3.2 is computed per pixel (or feature grid location h and w) or not, it is hard for me to grasp the idea of local distillation. I would suggest that the authors provide a figure for better understanding, like the ones in https://arxiv.org/pdf/2103.14030.pdf (Fig. 1). BTW, what is the “K” used in the experiments?
>
> **A.** It is computed per pixel. To clarify the process, we have added a new figure (Figure 2) with the dimension size in the proposed method. We changed the equation for explaining loss function using local maps. “K” is the number of local maps from splitting a global map. We used K = 4.
>
> **Q.** Overall, section 3 needs significant improvement. As a journal submission, the authors should provide more details such that the readers can reproduce the results. I would also suggest that the authors list the objective functions of highly relevant baseline methods.
>
> **A.** We reorganized the section on “Proposed Method”. Also, we explained in more detail in the section with modified equations and added a new figure (Figure 2).
>
> ***
> **W3**
>
> **Q.** Since all the compared methods more or less follow Eq (1), I would suggest that the authors clearly list down all the compared methods’ objective functions to provide a detailed comparison.
>
> **A.** In this paper, as we explained in the experiment section, all methods are implemented with the standard KD and their settings are all the same with our proposed method to see if the method can complement.
>
> **Q.** What is the purpose of section 4.6.3? It went across 2~3 pages (P 14-16) but I can hardly understand its purpose or importance.
>
> **A.** Generally speaking, even if a model shows a good accuracy in classification, it could generate miscalibrated performance. Or, a given method cannot always combine with other existing robust methods to produce positive results. To evaluate generalizability of each method and to see if it can combine it with the other methods, we utilized various existing robust methods, not only augmentation but also feature distillation and technique for masking. As shown in some results, there are some cases that other methods cannot complement existing methods, but ours can very well combine with it. Thus, the purpose of this section is to show that our method has the benefit that it can combine with various existing methods to complement and obtain better results.
>
> ***
>
> **W4**
>
> **Q.** The authors only conducted experiments on small-scale datasets. Even the paper by Sergey Zagoruyko and Nikos Komodakis in 2017 has experimented on ImageNet, CUB, and Scenes. The authors should consider more datasets in terms of their scale and diversity to justify the effectiveness of the proposed method.
>
> **A.** We have now added the results on ImageNet-1k in Table 6. AMD(global) shows better results than other methods we explored, increasing the top-1 and top-5 accuracy by 1.83% and 1.43% over the results of learning from scratch, respectively. The experimental setting is explained in section 5.1.2.
>
> **Q.** The teacher networks seem not very strong (e.g., ResNet44). I would suggest that the authors consider stronger teacher models, like the ones studied in https://arxiv.org/pdf/2006.10029.pdf
>
> **A.** Because of computational complexity and lack of resources, we implemented some small sized teachers but our experiments still include large sized teacher models (WRN16-8, WRN28-3, WRN40-2) and their compression ratio.

---

> ### Author Response · Authors · 2022-05-18
> **Response to Reviewer ALGP [Part3]**
>
> **Q.** The improvement is quite small. In many of the settings, the improvement over the baselines is < 1.
>
> **A.** We implemented a standard KD for all baselines so the difference of results could be small but we ran it sufficiently many times and reported with averaged accuracy with standard deviation. To present additional results, we have added experimental results on ImageNet-1k in Table 6. In ImageNet experiments, the proposed method improved the performance by 1.83% compared to the model trained from scratch. We do however note that, the difference between accuracies of methods could be small, but we ran sufficiently many times and reported the averaged accuracy. Also, marginal difference is commonly reported in prior works [1, 2], even with improvement sometimes less than 0.1% [1,2,3,4], since distillation is more directly concerned with model compression.
>
> [1] Sergey Zagoruyko and Nikos Komodakis. Paying more attention to attention: Improving the performance of convolutional neural networks via attention transfer. In Proceedings of the International Conference on Learning and Representations, pp. 1–13, 2017. \
> [2] Jang Hyun Cho and Bharath Hariharan. On the efficacy of knowledge distillation. In Proceedings of the IEEE/CVF International Conference on Computer Vision, pp. 4794–4802, 2019. \
> [3] Mingi Ji, Byeongho Heo, and Sungrae Park. Show, attend and distill: Knowledge distillation via attention-based feature matching. In Proceedings of the AAAI Conference on Artificial Intelligence, volume 35, pp. 7945–7952, 2021. \
> [4] Yonglong Tian, Dilip Krishnan, and Phillip Isola. Contrastive representation distillation. arXiv preprint arXiv:1910.10699, 2019.
>
>
> **Q.** If possible, more baseline methods should be compared. For now, only AFDS and AFD are proposed after 2020. Given that knowledge distillation is a widely studied subfield in ML, I would expect that more recent baselines are involved. Or, at least the authors should provide a reason why more recent methods are not compared.
>
> **A.** The proposed method is basically based on attention map distillation so that we mainly selected feature distillation based methods and specially using attention map based methods, explained in section 5.2.
>
> ***
>
> **W5**
>
> **Q.** The related work is well-written but can contain some more works. For example, several compared baselines (e.g., Yim et al.) are not discussed in the related work. Besides, I would like the authors to clearly position their work in the context of the related work. . For example, what is the difference between the proposed method and related work?
>
> **A.** The proposed method is basically based on attention map distillation so that we mainly selected feature distillation based methods and specially using attention map based methods. To explain more details about our method and differences from previous works, we added a new figure (Figure 2) and reorganized the section for explaining the proposed method.

---

### Decision · Action_Editors · 2022-06-08

**Recommendation:** Reject

**Comment:**

The paper are reviewed by three expert reviewers in the field. While the reviewers appreciate the extensive analyses and ablation studies in the paper, overall the reviewers have significant concerns over several aspects of the paper.

- Insufficient technical novelty (e.g., angular similarity is widely used)
- Unclear method exposition (particularly for Sec 3)
- Limited set of experiments (e.g., lack of larger-scale datasets, missing comparisons with more recent methods).

The authors' response partially addressed the reviewers' concerns, e.g., revising the writing of Sec 3 and providing the results on ImageNet-1K. Unfortunately, these response did not fully convince the reviewers. After reading the responses, all three reviewers recommended "Leaning Reject". The AC read the reviews, responses, and checked the paper revision. The AC agrees with the reviewers that the amount of required changes do not justify the Accept with minor revision decision (i.e., the paper at the current state should go through another round of major revision and resubmit for review). The AC thus recommends to reject.